# Defective oxygen inert phase stabilized high-voltage nickel-rich cathode for high-energy lithium-ion batteries

Zhongsheng Dai[1], Zhujie Li[2], Renjie Chen [1,2,3] ✉, Feng Wu[1,2,3] & Li Li [1,2,3] ✉

Pushing layered cathode to higher operating voltage can facilitate the realization of high-energy lithium-ion batteries. However, the released oxygen species initiate materials surface upon highly delithiated states will react severely with electrolyte, accelerating the structure deterioration and triggering the thermal degradation. Here we propose an inert phase of $La_2Mo_2O_9$ with abundant oxygen vacancies (about 41%) by regulating the annealing temperature to engineer the cathode interface beyond conventional modifications. By employing $LiNi_{0.8}Co_{0.1}Mn_{0.1}O_2$ as a model system and extending to higher voltage-operated $LiCoO_2$ and Li-rich cathode, we demonstrate that the introduced lanthanum and molybdenum ions will transfer electrons to enhance the surface oxygen electronegativities, thus served as "oxygen anchor" to alleviate oxygen evolution. Furthermore, the possible released oxygen can be operando captured and reserved by β-phase $La_2Mo_2O_9$ depositor for the intrinsic high oxygen vacancy formation energy. The reaction involving oxygen species with electrolyte is fundamentally diminished, thus effectively mitigate the structure deterioration and elevate the electrochemical performances, enabling a 1.5-Ah pouch-type full cell to exhibit negligible 6.0% capacity loss after 400 cycles.

Increasing the operating voltage of the layered cathode could elevate the energy density for lithium-ion batteries (LIBs)[1]. However, the enhanced cut-off voltage results in large-scale degradation from the material surface to the bulk[2]. When the gradually lowered Femi level $E_F$ of the cathode contacts with the band of oxygen upon charging, charge transfer might occur between the $3d$ electrons of transition metal ions (TMs) and the $2p$ electrons of oxygen, leading to the evolution of oxygen species ($O_2$, $O^{2-}$, $O^-$)[3]. The liberation of oxygen species is believed to initiate on the cathode surface and would react speedily with electrolyte, bringing in the phase transition and the generation of cathode-electrolyte interface (CEI), which impeded the Li-ion diffusion kinetics[4]. Even worse, this reaction could generate a large amount of heat to trigger serious thermal runaway[5]. Furthermore, the dramatic shrinkage of lattice $c$ in the high state of charge (SOC) is also deemed to

be the charge transfer between TMs and oxygen, which could incur the generation of lattice strain and microcracks, exposing more particles to electrolyte and intensifying the gas release and structure instability[6,7].

Substantial efforts have been made to restrain the oxygen evolution originating from the materials surface and improve the electrochemical performance of layered cathode, including surface coating and heteroatoms doping[8,9]. The atoms of coating layer might form chemical bonds with surface oxygen to improve the stability. Simultaneously, the coating layer could suppress the surface reactions and alleviate the stress accumulation[10]. Incorporating heteroatoms with high binding energy to lattice oxygen would regulate the $2p$ band and introduce negative charge for oxygen atoms, thus validly inhibiting the evolution[11]. However, these commonly used strategies are

[1]Beijing Key Laboratory of Environmental Science and Engineering, School of Materials Science and Engineering, Beijing Institute of Technology, Beijing 100081, P. R. China. [2]Advanced Technology Research Institute, Beijing Institute of Technology, Jinan 250300, P. R. China. [3]Collaborative Innovation Center of Electric Vehicles in Beijing, Beijing 100081, P. R. China. ✉e-mail: chenrj@bit.edu.cn; lily863@bit.edu.cn

indeed weakened after perennial cycles. Consequently, finding a suitable method to modulate the oxygen species evolution is essential for achieving safe and sustainable high-energy density LIBs. Considering the sufficient intrinsic oxygen vacancies in solid oxide fuel cell electrolyte[12], we propose a robust structure different from the common ones, wherein depositing an oxygen defective layer on the materials surface. Nevertheless, due to the ubiquitous fast transport characteristic of oxygen ion, an inert phase with sufficient oxygen defects and dilatory ions dynamics is indispensable.

We employ the Ni-rich cathode $LiNi_{0.8}Co_{0.1}Mn_{0.1}O_2$ (P-NCM) with great industrial interest to illustrate the modification strategy. Furthermore, a lanthanum molybdate ($La_2Mo_2O_9$: LMO) depositor was selected based on the following aspects: (1) due to the high binding energy of lanthanum (La) and molybdenum (Mo) ions with oxygen, the electrons would be transferred from La and Mo to oxygen, thus it could serve as an "oxygen anchor" to stabilize surface oxygen; (2) an ~41% inherent oxygen vacancies in LMO would effectively capture and reserve oxygen species for the surface oxygen not anchored by La and Mo; (3) the phase transition temperature between inert (β) and active (α) LMO is 580 °C[13], well below the synthetic temperature of P-NCM, eliminating the effect of high-temperature operation on the bulk material. To prove the operando anchoring and capturing process, comprehensive investigations have been employed. Remarkably, the LMO-modified NCM811 (L-NCM) exhibits dramatically improved cycling stabilities under different cut-off voltages in half cells. A 1.5-Ah pouch cell composed of L-NCM cathode and commercial graphite (Gr) anode achieves an extraordinary capacity retention of 94% after 400 cycles. Extending to higher-voltage $LiCoO_2$ and Li-rich cathode, LMO modification could still effectively improve the electrochemical performances.

## Results

### Operando oxygen anchoring and capturing strategy

Figure 1a unambiguously illustrated the above-discussed failure mechanism of Ni-rich material and the operando anchoring and capturing strategy for oxygen. The selected area of X-ray diffraction (XRD) result confirmed that the β-phase for pure LMO was obtained under 500 °C (Supplementary Fig. 1)[13]. Therefore, the heat treatment of L-NCM was also in this condition. Clearly, the XRD patterns and corresponding Rietveld refinement results (Supplementary Figs. 2–3, Fig. 1b) manifested that both P-NCM and L-NCM were indexed to the R-3m space. The almost unchanged lattice parameters (Supplementary Table 1) indicated that the heteroatoms were not incorporated into L-NCM structure. Scanning electron microscopy (SEM) and energy dispersive spectra (EDS) mapping images revealed that the surface of L-NCM became ambiguous (Supplementary Fig. 4b) and La, Mo elements were enriched (Supplementary Fig. 5). Specially, cross-section morphology of L-NCM obtained by focused ion beam etching combined with SEM (FIB-SEM) and the mapping images (Supplementary Fig. 6) demonstrated that La and Mo elements were uniformly distributed on the material surface, illuminating the homogeneous LMO surface deposition.

The high-resolution transmission electron microscopy (HR-TEM) image (Fig. 1c, left) demonstrated that the lattice fringes with an average space of ~0.20 nm was corresponding to the (104) plane within bulk L-NCM, and the heterogeneous lattice fringes with a distance of ~0.23 nm was ascribed to the (220) plane of LMO. Furthermore, EDS mapping (Fig. 1c, right) from another perspective manifested the LMO deposition. The well-preserved layer structure near the L-NCM surface in the high-angle annular dark-field scanning transmission electron microscopy (HAADF-STEM) image (Fig. 1d) confirmed that the LMO alteration had no impact on the bulk structure. Particularly, more oxygen vacancies were detected in L-NCM surface (Supplementary Fig. 7c) on the X-ray photoelectron spectroscopy (XPS) spectra[14], which might originate from the LMO (enriched oxygen defect) layer.

The decreased impurity oxygen for L-NCM would effectivity alleviate the slurry flocculation during the mixing process[15], which could be manifested by the residual lithium content detect (Supplementary Fig. 8).

The high surface-sensitive O K-edge X-ray absorption spectroscopy (s-XAS) with total electron yield (TEY) mode could also provide the information of oxygen 2p orbital[16]. As shown in Supplementary Fig. 9, the enhanced pre-edge (~529 eV) in uncharged L-NCM corresponding to the decreased charge density around the oxygen ion, illustrating more oxygen vacancies were appeared in L-NCM surface[17], which could be attributed to the oxygen defect in LMO layer. Furthermore, the high relative intensity of L-NCM electron paramagnetic resonance (EPR) spectroscopy (Fig. 1e) confirmed that the total oxygen vacancies were higher than their pristine counterpart. In addition, the slightly decreased $Ni^{2+}$ near L-NCM surface (Supplementary Fig. 10) could induce the mitigated $Li^+/Ni^{2+}$ cations disorder, coinciding with the XPS results.

### Moderated oxygen activity

As shown in Fig. 1f, obvious oxygen release could be monitored for P-NCM in the in situ differential electrochemical mass spectrometry (DEMS) characterization, whereas no gas evolution was involved in L-NCM (Fig. 1g), demonstrating that the irreversible oxygen redox was effectively mitigated under LMO treatment. Actually, the electrochemical activity of lattice oxygen was directly connected with the charge compensation. Especially in high SOC, the bottom of TMs 3d band might be in contact with the oxygen 2p band, resulting in the electrons transportation from oxygen and thus the gas liberation[18]. Consequently, reduced oxygen escape in L-NCM might be attributed to that the electrons transferred from La and Mo ions to the bonded surface oxygen, which elevated the oxygen electronegativities to resist exorbitant oxidation upon high-voltage charge compensation (oxygen anchor). Furthermore, the possible released oxygen could be operando captured and reserved by LMO shielding layer with abundant oxygen vacancies.

To elucidate the alleviated electron transfer between TMs and surface oxygen, the s-XAS of Ni $L_3$-edge with TEY mode at 1st fully charged state (4.5 V, Fig. 1h) were detected. Based on the standard spectral of NiO, $Ni_2O_3$ and $NiO_2$, the $Ni^{3+}$ and $Ni^{4+}$ were decreased in P-NCM sample surface, whereas the content of $Ni^{2+}$ was elevated compared with their counterpart, manifesting that the charge transfer in L-NCM surface were mitigated[16]. In addition, the reduction of $Ni^{4+}$ in P-NCM surface could ascribe to the detrimental redox reactions with electrolyte was happened. The O K-edge s-XAS spectra for the 1st charged state (Supplementary Fig. 11a) confirmed that the TMs 3d-oxygen 2p rehybridization in P-NCM was elevated, whereas the pre-edge in L-NCM was almost unchanged, demonstrating that the more oxygen vacancies were generated in P-NCM surface and the oxygen activities were mitigated in L-NCM, coinciding with the in situ DEMS analysis, which could be testified by the EPR tests (Supplementary Fig. 11b).

To ascertain the durable effect of LMO compound on restraining gas evolutions, the electrodes after 200 cycles in coin cells were disassembled and investigated by in situ DEMS test (Supplementary Fig. 12). The slighter oxygen escape in modified sample compared with P-NCM provided solid evidence that LMO depositor decreased the oxygen activity. Furthermore, the severely enhanced O s-XAS pre-edge and the relative intensity of EPR profile for 200th charged P-NCM sample illustrated abundant oxygen vacancies were formed (Supplementary Fig. 13). Nevertheless, the intensities in L-NCM were slightly enhanced compared with the uncharged state, confirming the decreased oxygen activities after LMO modification.

In situ heating XRD (HT-XRD) analysis was employed to assess the phase transition for the delithiated cathodes under harsh thermal conditions (Fig. 1i–k). The layered structure (R-3m) of P-NCM was

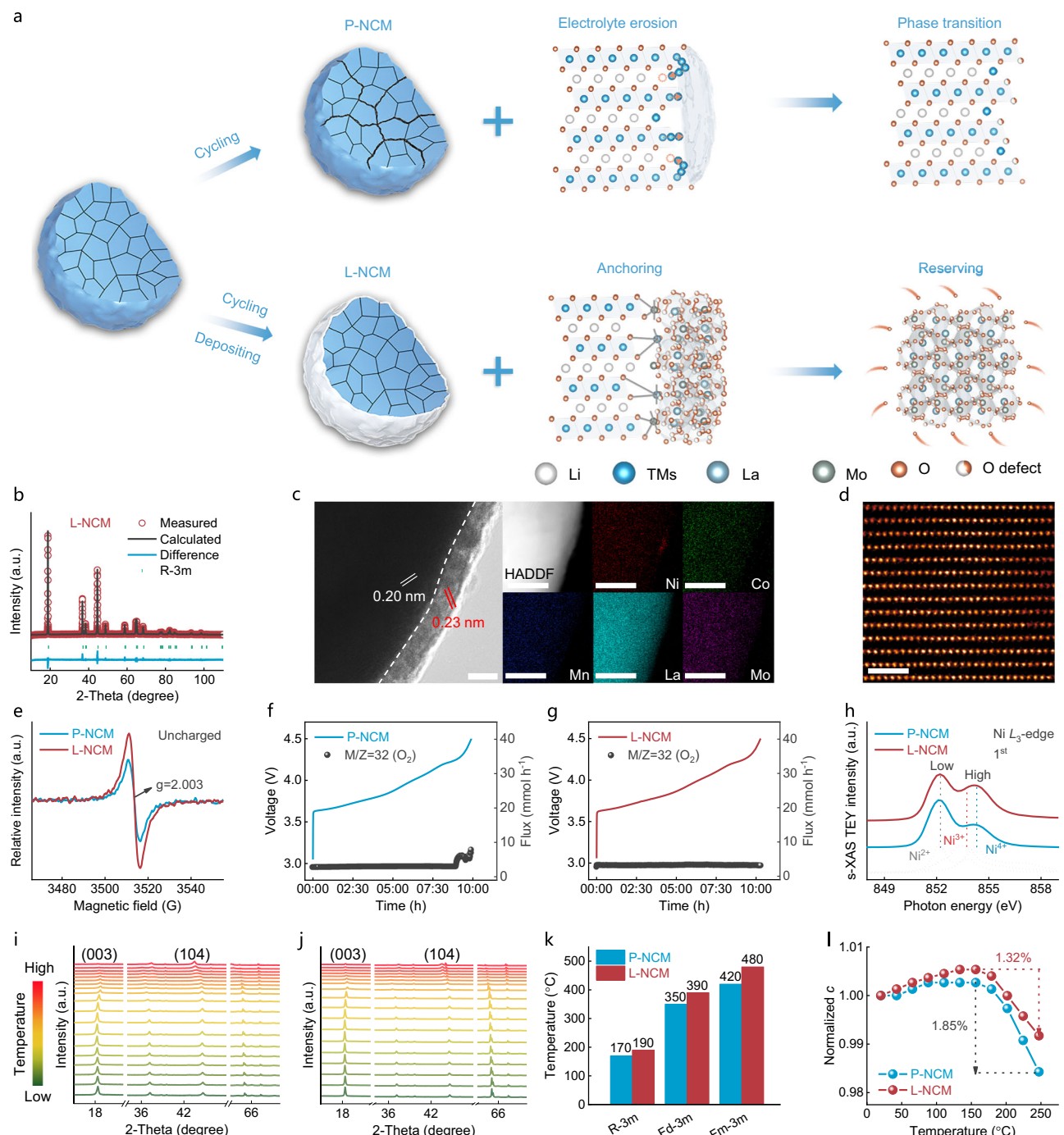

**Fig. 1 | Oxygen anchoring and reserving strategy. a** Schematic illustrations of operando oxygen anchoring and reserving strategy. **b** XRD Rietveld refinement for uncycled L-NCM. **c** HR-TEM (scale bars, 5 nm) and elemental mapping images (scale bars, 50 nm) for uncycled L-NCM. **d** HAADF-STEM image for uncycled L-NCM. Scale bars, 1 nm. **e** EPR profile at uncharged state. **f, g** In situ DEMS data for P-NCM (**f**) and L-NCM (**g**) during 1st charge cycle under 2.7–4.5 V. **h** s-XAS spectra of Ni $L_3$-edge at 4.5 V in 1st charged state. **i, j** In situ HT-XRD patterns for delithiated P-NCM (**i**) and L-NCM (**j**) cathodes at 4.5 V in the temperature range of 25–600 °C. **k** Phase transformation temperature. **l** Evolutions of normalized $c$-axis upon heating.

transformed into a disordered spinel phase (Fd-3m) when the temperature elevated to 170 °C and then completely transformed into rock-salt phase (Fm-3m) at 350 °C. By contrast, the phase transformation temperature of layered L-NCM to disorder structure was increased to 190 °C and the spinel phase remained well even heating up to 390 °C. The phase transition of Ni-rich cathode upon heating could be ascribed to the lowered valence TMs (principally for $Ni^{2+}$) spontaneously migrated to the Li layer[19]. This process was also related to the charge transfer between TMs and oxygen, thus the elevated phase transformation temperature for L-NCM electrode validly

confirmed the lowered oxygen activities and the enhanced thermal stabilities. The lattice parameter variations upon heating were further quantified by XRD Rietveld refinement (Fig. 1l, Supplementary Fig. 14). The more obvious $c$-axis expansion under moderate temperature and slightly decline at the elevated temperature in L-NCM sample ascertained that the structure was more stable than P-NCM.

## Elevated electrochemical performances

Figure 2a exhibited that the initial specific capacity of L-NCM was similar to P-NCM. Differential capacity (dQ dV$^{-1}$) profiles of 1st cycle

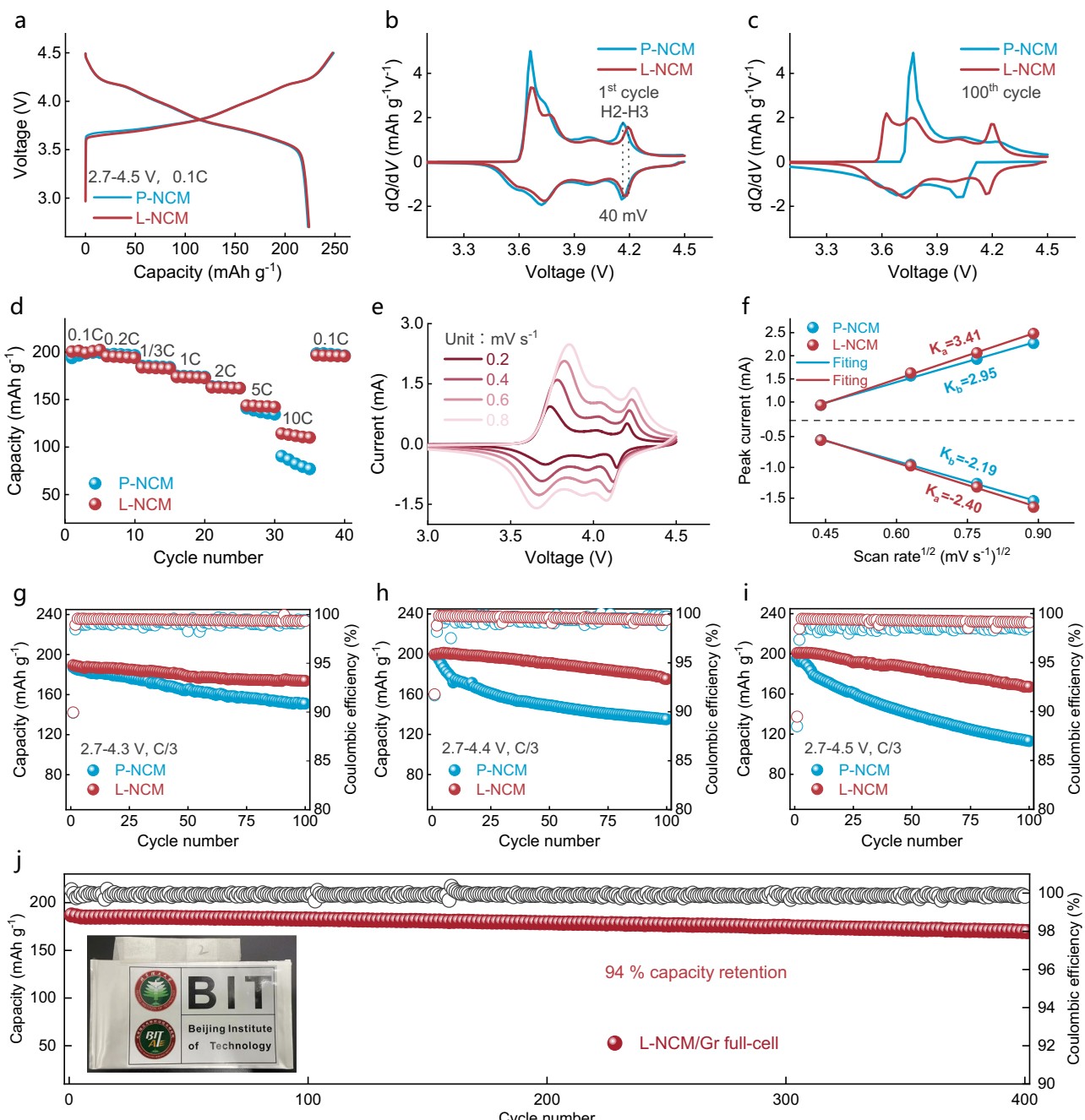

**Fig. 2 | Electrochemical performances for coin/pouch-type cells. a** Initial charge-discharge curves of coin-type half cells in 0.1 C under 2.7–4.5 V. **b, c** dQ dV$^{-1}$ profiles of 1st cycle (**b**) and 100th cycle (**c**) in coin-type half cells under 2.7–4.5 V. **d** Rate capability in coin-type half cells under 2.7–4.5 V. **e** CV curves for L-NCM in coin-type half cell. **f** Linear relationship fitting of I$_p$–v$^{1/2}$. **g–i** Cycling performances of coin-type half cells containing P-LNCM and L-NCM, under 2.7–4.3 V (**g**), 2.7–4.4 V (**h**) and 2.7–4.5 V (**i**) at the current of C/3. **j** Cycling performance of L-NCM/Gr pouch-type full cell at 0.5 C under 2.7–4.2 V. Inset: photo of the assembled pouch cell (size: 5 cm long and 8 cm wide).

(Fig. 2b) showed that the H2-H3 redox peak shifted to a higher potential for L-NCM, implying the abrupt lattice change might be postponed. After 100 cycles, the intensity of H2-H3 peak in L-NCM was remained (Fig. 2c), whereas the reversibility in P-NCM was mitigated, manifesting the severely structure/capacity destruction upon cycling[2]. The modified cathode also exhibited a desirable rate capability compared to P-NCM (Fig. 2d), illustrating the improved Li-ion transport dynamics (Fig. 2e, f, Supplementary Fig. 15, Table 2) and decreased surface charge transfer resistance (Supplementary Fig. 16). As revealed in Fig. 2g–i and Supplementary Table 3, the electrochemical properties of L-NCM were obviously improved at different cut-off voltages, and

were found superior compared with the recent similar-focused studies (Supplementary Table 4).

Particularly, the LiCoO$_2$ and Mn-based Li-rich cathode could achieve a desirable discharge capacity through elevating the operating potential, whereas grievous oxygen release was also involved. Therefore, higher-voltage LiCoO$_2$ and Li-rich cathode were applied to further verify the special oxygen release suppression effect of LMO deposition. The results shown in Supplementary Fig. 17 provided solid evidence that the LMO modification could also enhance the cycling durability. In view of the excellent electrochemical performance of L-NCM in half cells, a 1.5-Ah pouch-type full cell composed of L-NCM

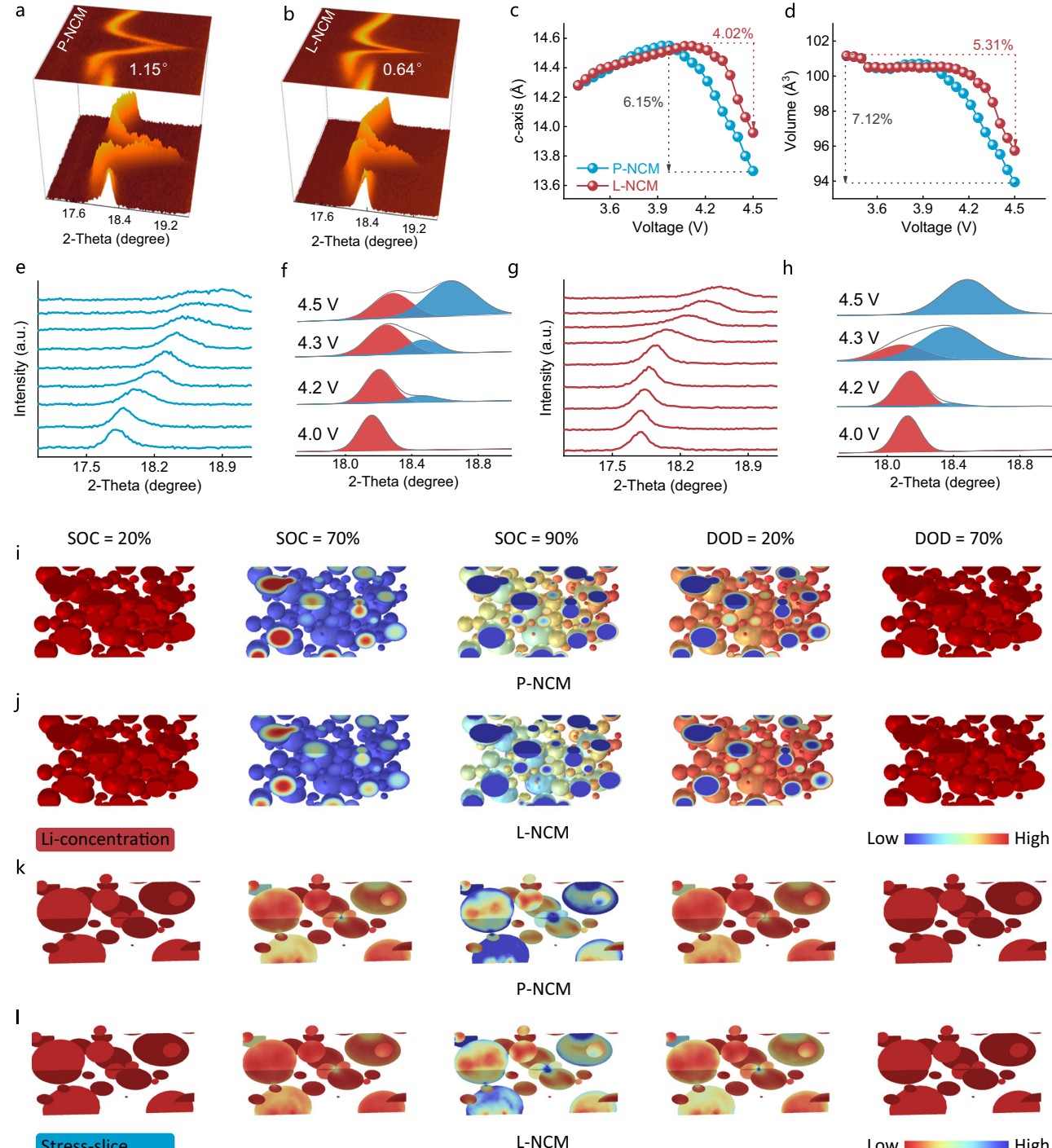

**Fig. 3 | Crystal structure, Li-ion concentration, and stress evolution.**
**a**, **b** Selected (003) counter plot of in situ XRD patterns for P-NCM (**a**) and L-NCM (**b**) upon initial cycle. **c**, **d** Lattice parameter variations for *c*-axis (**c**) and volume (**d**) obtained by quantified the in situ XRD data. **e**–**h** In situ XRD curves of (003) peak and corresponding deconvolution between 4.0 V and 4.5 V for P-NCM (**e**, **f**) and L-NCM (**g**, **h**). **i**, **j** Li-ion concentration distribution for P-NCM (**i**) and L-NCM (**j**). **k**, **l** Stress field analysis for P-NCM (**k**) and L-NCM (**l**).

cathode and commercial Gr anode were assembled. The pouch cell achieved a reversible capacity of 170.5 mAh g⁻¹ after 400 cycles at 0.5 C (Fig. 2j), corresponding to a capacity retention of 94% and a coulombic efficiency of above 99.9%. The voltage decline was only 20 mV with a retention of 99% after 400 cycles (Supplementary Fig. 18). Consequently, the inert phase LMO treatment could validly elevate the electrochemical performance of layered cathode with an eliminated oxygen species evolution and surface side reaction, even under ultra-high voltage conditions.

## Structure detection and multi-physical simulation
In situ XRD characterization was conducted to track the structure change upon initial cycling (Fig. 3a, b, Supplementary Fig. 19), and the lattice parameters were quantified by the refinement XRD patterns (Fig. 3c, d, Supplementary Fig. 20). The (003) peak shift above 4.18 V for P-NCM and L-NCM was 1.15° and 0.64°, respectively, corresponding to the maximal *c*-axis contraction of 6.15% and 4.02% under 4.5 V cut-off voltage. Furthermore, the volume variations for P-NCM and L-NCM were 7.12% and 3.31%, respectively. The shrinkage of lattice parameter *c*

at high SOC was primarily ascribed to the high valence TMs received electrons from the coordinated oxygen[19,20], resulting in the oxidation of oxygen ion and thus the repulsive force was weakened in delithiated Li layer. As shown in Supplementary Fig. 21, the maximum shrinkage of lattice parameter $c$ for L-NCM was 0.4% under 4.3 V, much smaller than the pristine counterpart (2.8%), which helped the modified material approaching to zero-strain, thus effectively mitigated the seriously grain boundary side reactions.

The enhanced structure stability of L-NCM might be ascribed to the surface electrons transition from La and Mo ions to the coordinate oxygen, elevating the oxygen electronegativities and thus mitigating TMs 3$d$-oxygen 2$p$ orbital overlap at highly delithiated state, which eventually alleviated the lattice collapse resulted by the oxygen release from surface to bulk. Figure 3e–h illuminated the (003) profiles and the corresponding deconvolution between 4.0 V and 4.5 V. The deconvoluted (003) peak for P-NCM exhibited two coexisting phases till 4.5 V, whereas the phase transition should be finished at ~4.2 V according to the dQ dV$^{-1}$ curves. Comparatively, slight H2 phase could be observed (4.3 V) and the phase transformation was fully completed with increased charging voltage within the modified sample. The improved phase transition dynamics could generate homogenies Li-ion concentration and the consequently uniformed strain distribution in L-NCM[21].

Multi-physical simulation was performed to visualize the Li-ion concentration and the corresponding stress field for both electrodes based on the difference of ionic diffusion coefficient. An evidently heterogeneous concentration distribution could be observed in P-NCM (Fig. 3i) even at highly delithiated state (90%), partial Li-ion was still retained on the cathode surface, which could be attributed to the limited Li-ion mass-transfer caused by the intrinsic inferior ionic diffusion coefficient[22]. Nevertheless, the uniformed Li-ion concentration was achieved in the modified sample due to the improved diffusion kinetics, especially in 90% SOC, almost non-disparity inside particles and the surface (Fig. 3j). The homogeneous Li-ion concentration field would mitigate the strain generation and equilibrate the distribution of stress field as indicated in Fig. 3k, l.

## Surface reconstruction and side reaction for electrode

Time-of-flight secondary-ion mass spectrometry (TOF-SIMS) analysis was carried out to ascertain the surface compositions and the depth distributions of cathode interphase for the 200 cycled cathodes (Fig. 4a–f, Supplementary Fig. 22). As indicated in Fig. 4a–d, the normalized intensity for the select fragments demonstrated that the CEI film induced by surface side reactions were mitigated in L-NCM sample compared with its pristine counterpart[23]. The 3D reconstruction and TOF-SIMS chemical images (Fig. 4e, f) manifested the thinner gradient concentration of interface degradation-generated species within the modified cathode, confirming that the electrolyte decomposition reactions were effectively alleviated. Those results have been further examined by XPS test on the cycled cathode. Particularly, the obviously weakened OCO$_2$ and LiF peak intensities in L-NCM surface demonstrated the reduced electrolyte decomposition (Supplementary Fig. 23)[24,25].

Lithium metal batteries with higher energy density have drawn extensive studies and the TMs dissolution from cathode would significantly deteriorate the electrochemical performance of lithium metal anode (LMA)[26]. Clearly, the LMA surface appeared NiF$_2$ and NiO peaks after 200 cycles in the depth XPS profile (Supplementary Fig. 24). With the increasing etching time, the characteristic Ni-relevant signals were reduced in LMA near-surface coupled with L-NCM compared with those paired with P-NCM, illuminating the alleviated TMs decomposition after LMO deposition. As displayed in Fig. 4g (left), the rough interface of LMA coupled with P-NCM could be involved in SEM image, with an asperous layer composed of solid-electrolyte interphase (SEI)

and dead Li compound (Fig. 4g, right)[27]. Nevertheless, the LMA interface was smooth for those paired with L-NCM (Fig. 4h), and only an intact SEI layer was observed in the cross-section image[3]. EDS mapping profiles for the asperously layer (Fig. 4i, j) confirmed that the signals of TMs in LMA surface paired with L-NCM were weaker than the P-NCM, from another aspect elucidating the mitigated surface side reactions.

X-ray absorption near edge structure (XANES) technique with high energy hard X-ray could facilely penetrate the Ni-rich materials[28], thus could be employed to investigate the overall Ni valence state and the coordination environment (Fig. 5a, b). The Ni-$K$ edge for fully charged P-NCM was shifted to a lower energy position compared with the modified electrode after 100 and 200 cycles, suggesting that the overall valence of Ni ion was decreased, which could be explained by the spontaneous Ni$^{3+/4+}$ reduction[20]. The local chemical of Ni-$K$ edge has been accessed by the Fourier-transformed extended X-ray absorption fine structures (FT-EXAFS). The results shown in Fig. 5c, d manifested that both the Ni-O and Ni-TM (Ni, Co, Mn) coordination distances for P-NCM were lengthened after identical cycles, confirming the electrons transfer in highly charged state[29]. In contrast, the interatomic peak variations were almost negligible in cycled L-NCM, testifying highly stable local coordination environments. Wavelet-transformed EXAFS analyzed in Fig. 5e, f presented that the Ni-O bond was prolonged to the high-$\mathbf{k}$ direction in P-NCM, whereas no evident changes has been observed in the modified electrode, illustrating that the oxygen-related defect was fundamentally reduced.

Supplementary Fig. 25 shows that both samples maintained the initial layered structure, and the slighter (003) peak shift to a lower angle in L-NCM manifested the mitigation of Li deficiency and electrode polarization[30]. Furthermore, the morphological characterizations (Fig. 5g, h, Supplementary Figs. 26–27) exhibited the secondary particles of P-NCM electrode experienced seriously broken, whereas the intracrystalline microcracks were reduced after LMO treatment, providing solid evidence for elevated structure stabilities. The stress accumulation for cycled cathodes were further investigated by finite element analysis and the models were established based on the cross-section SEM images (Fig. 5i, j). Obviously, the larger detrimental stress distribution has been observed in the local area of P-NCM primary particles, which would destruct the mechanical integrity of the material and expose more particles surface to electrolyte, accelerating the oxygen release and eventually deteriorating the electrochemical performance.

The clearly intragranular microcracks could be observed in 200 cycled P-NCM primary particle surface (Supplementary Fig. 28a), illustrating the severe stress generation. However, the microstructure of modified sample was fairly remained with superior mechanical integrity (Supplementary Fig. 28b). As exhibited for the near-surface HAADF-STEM image (Fig. 5k), three distinct regions could be observed in P-NCM. Apart from the original layered structure inside the bulk (region I), P-NCM presented a reconstructed surface with a thickness of ~10 nm, which could be identified to NiO-like rock-salt phase and confirmed by the fast Fourier transform patterns (FFT)[31]. The disordered layer was propagated towards the inner bulk for ~15 nm and it would be indexed to a spinel structure. In contrast, L-NCM sample maintained the characteristic layered structure and only ~3 nm disordered thin layer was monitored on the near-surface (Fig. 5l). Furthermore, the intact LMO shielding layer with P123 space group was tightly attached to its surface and proved by FFT[32]. The improved surface electrochemical environment in the modified electrode could be attributed to the defective oxygen inert LMO depositor, which not only validly stabilized the surface oxygen species by impeding the charge transfer between TMs and oxygens, but also alleviated interfacial side reactions caused by the high valence TMs with electrolyte.

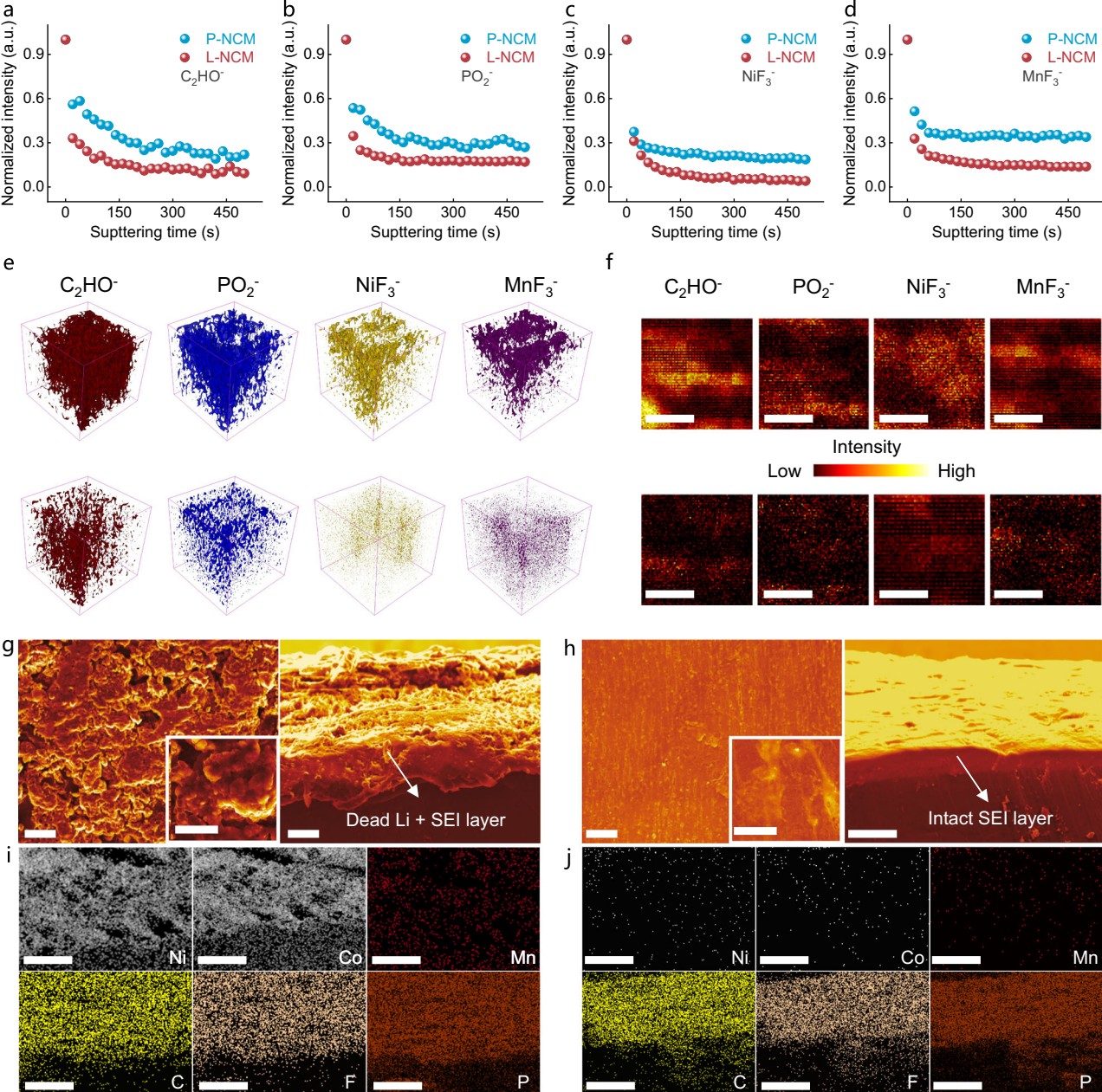

**Fig. 4 | Cathode, and anode surface chemistry analysis. a–d** TOF-SIMS depth profiles of $PO_2^-$ (**a**), $C_2HO^-$ (**b**), $NiF_3^-$ (**c**) and $MnF_3^-$ (**d**) for P-NCM and L-NCM after 200 cycles. **e** 3D rendering TOF-SIMS fragments. **f** Surface species distributions. Scale bars, 5 μm. **g** Top-view (left, scale bars, 20 μm) and cross-section morphology (right, scale bars, 10 μm) images of LMA after 200 cycles coupled with P-NCM.

**h** Top-view (left, scale bars, 20 μm) and cross-section morphology (right, scale bars, 10 μm) images of LMA after 200 cycles coupled with L-NCM. **i, j** Cross-section elemental mapping images of LMA after 200 cycles, coupled with P-NCM (**i**) and coupled with L-NCM (**j**). Scale bars, 10 μm.

## Theory simulation on oxygen activity

Density functional theory (DFT) calculation was carried out to elucidate the oxygen evolutions in layered cathode and the operando anchoring and capturing mechanisms of LMO depositor. As $LiNiO_2$ was selected as a model, $NiO_6$ and $LiO_6$ octahedra were truncated on its (104) surface. Typically, two kinds of uncoordinated surface oxygen were involved, including type I: one oxygen bonded with two Ni and three Li; type II: one oxygen bonded with three Ni and two Li. As could be observed in Fig. 6a and Supplementary Fig. 29a, the Ni $3d$ and O $2p$ orbitals exhibited strong hybridization, which helped the occupied O $2p$ states (close to Fermi level $E_F$) to be more localization. Nevertheless, a lack of Li-O-Ni configuration promoted the discretization for surface oxygen (type I) near $E_F$, resulting in the weakened Ni $3d$ and O $2p$

hybridization (Fig. 6b, Supplementary Fig. 29b), which could be considered as the origin of surface oxygen instability and responsible for the charge compensation under high SOC, which contributed to the oxygen species evolutions[33].

The cations within coating layer were generally coordinated with the surface oxygen of bulk material, thus the bonding effect between La, Mo, and surface oxygen were analyzed. Due to the high binding energy of interface La-O and Mo-O compared with Ni-O, the energy states of surface oxygen coordinated with La and Mo were decreased (Fig. 6c, d, Supplementary Fig. 29c, d), contributing to less high-energy states approaching to $E_F$ than that of exposed surface oxygen (Fig. 6b) and lattice oxygen (Fig. 6a). The lowered energy level of surface oxygen after LMO deposition could be attributed to the charge transfer

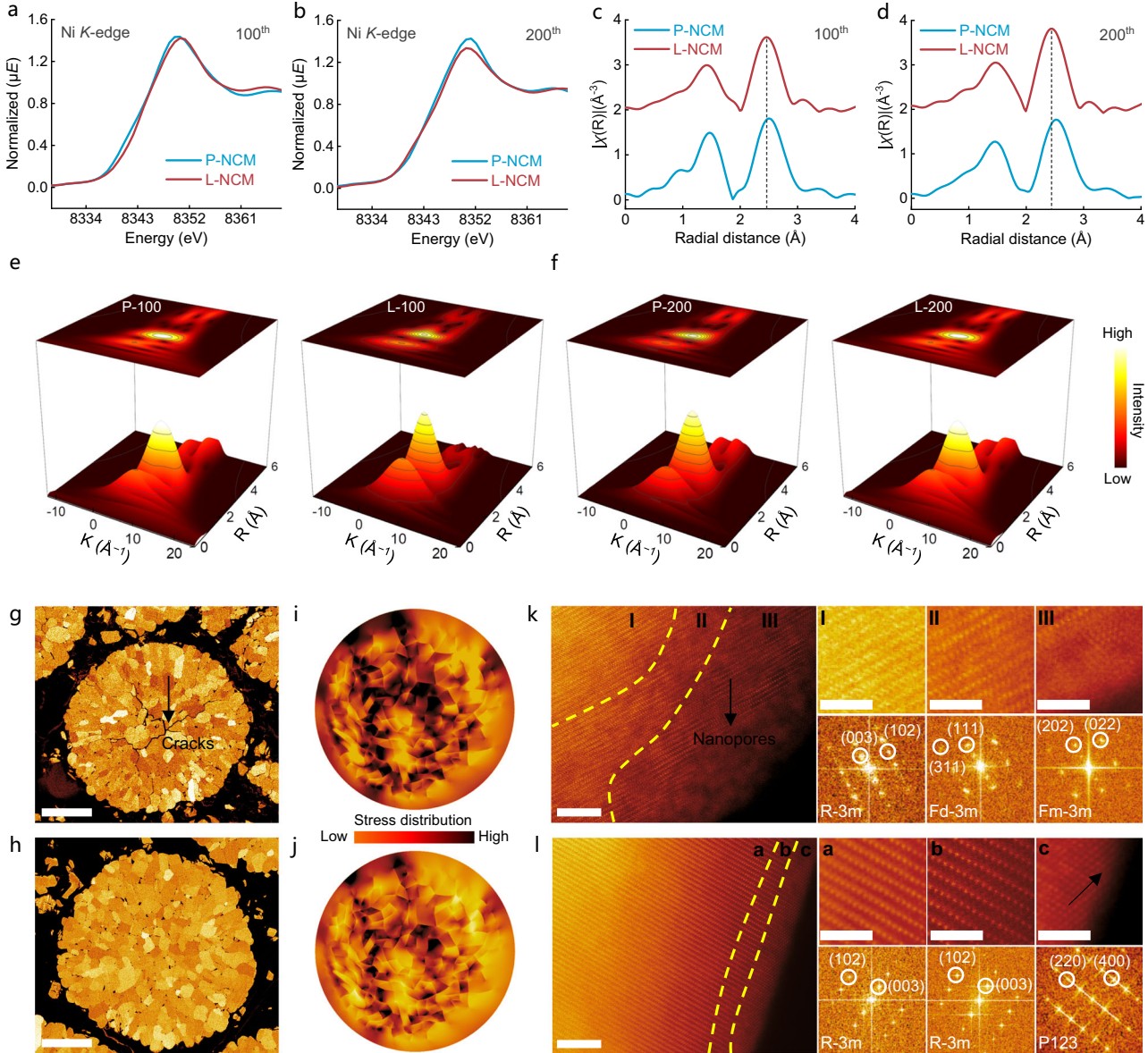

**Fig. 5 | Coordination environment, morphology, and microstructure investigations. a**, **b** Ni-*K* edge XANES spectra for 100th (**a**) and 200th (**b**) cycled electrodes. **c**, **d** FT-EXAFS spectra for 100th (**c**) and 200th (**d**) cycled electrodes. **e**, **f** Wavelet-transformed EXAFS for 100th (**e**) and 200th (**f**) cycled electrodes. **g**, **h** Cross-section SEM images for P-NCM (**g**) and L-NCM (**h**) after 200 cycles. Scale bars, 5 µm. **i**, **j** Stress distributions for P-NCM (**i**) and L-NCM (**j**) after 200 cycles (the mode based on the cross-section SEM image). **k**, **l** HAADF-STEM and FFT profiles for P-NCM (**k**) and L-NCM (**l**) after 200 cycles. Left, scale bars, 5 nm; right (enlarged area), scale bars, 2 nm.

from La and Mo ions, validly elevating the electronegativities/stabilities, which would be demonstrated by charge density distribution (Fig. 6e, f) and Bader charge analysis (Supplementary Table 5), thus La and Mo could be regarded as the "oxygen anchor" in Ni-rich cathode surface. To illustrate the "oxygen reserving" effect of LMO layer, the intrinsic oxygen vacancy formation energy of LNO (104) and LMO (001) surface were calculated and discussed (Fig. 6g, h and Supplementary Fig. 30a, b). When oxygen vacancy was formed on LiNiO$_2$ (104) surface, an electronic state composed of Ni 3*d* and O 2*p* was introduced near the bottom of conduction band, and the calculated oxygen vacancy formation energy was ~ 3.1 eV. In contrast, when oxygen vacancy was generated on the LMO (001) surface (Fig. 6i, j and Supplementary Fig. 30c, d), an intermediate electronic state composed of La and Mo 4*d* orbital was introduced between the conduction band and the valence band. The oxygen vacancy formation energy of LMO was ~5.1 eV, which was about 2 eV higher than the LiNiO$_2$, indicating that the generation of oxygen vacancy was more difficult in LMO. Therefore, LMO shielding layer would effectively reserve those oxygen species not anchored by La and Mo.

## Discussion

In this work, the defective oxygen inert phase LMO was engineered on Ni-rich cathode surface and demonstrated successful suppression of the irreversible interfacial deterioration and bulk strain caused by oxygen evolutions, which helped L-NCM electrode to exhibit an elevated cycling stabilities in the cut-off voltage from 4.3 to 4.5 V. Extending to higher voltage-operated LiCoO$_2$ (4.6 V) and Li-rich (4.8 V) materials, LMO deposition could also enhance the electrochemical performances. Particularly, a pouch-type full cell composed of L-NCM cathode and commercial Gr anode achieved a remarkable capacity retention of 94% after 400 cycles under 0.5 C. The improved cycling capability and structure stability have been extensively investigated and could be summarized in the following aspects: (1) the strong La-O and Mo-O bond lowered the 2*p* energy states of coordinated oxygen

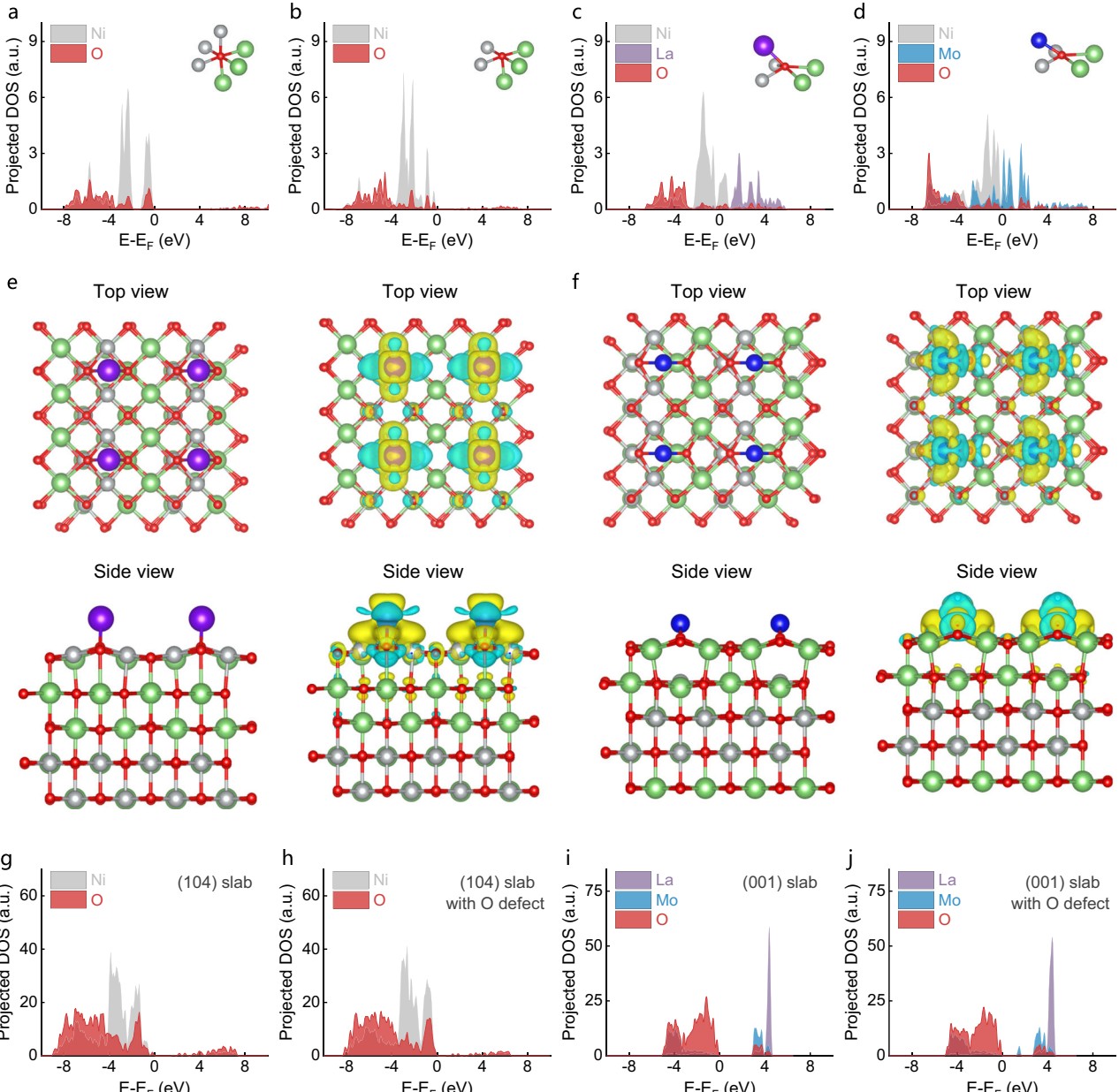

**Fig. 6 | Density functional theory calculations. a–d** Projected density of states (DOS) for lattice oxygen coordinated by three Li and three Ni (**a**), surface oxygen coordinated by three Li and two Ni (**b**), surface oxygen coordinated by one La, two Li and two Ni (**c**), surface oxygen coordinated by one Mo, two Li and two Ni (**d**). **e**, **f** Optimized structure (left) and charge density distribution (right) between LiNiO$_2$ surface and LMO, La bonded (**e**) and Mo bonded (**f**). **g–j** Projected DOS for LiNiO$_2$ (104) slab without defect (**g**) and with one oxygen defect (**h**), LMO (001) slab without defect (**i**) and with one oxygen defect (**j**). Note the elements: Li (green), Ni (gray), O (red), La (purple) and Mo (blue).

and thus served as "oxygen anchor" to alleviate the material failure caused by oxygen release originated from the surface; (2) the crystal structure was effectively stabilized due to the spontaneous charge transfer between TMs and oxygen was mitigated at highly delithiated cathode; (3) the β-phase LMO layer with 41% oxygen vacancies could be in situ captured and reserved the possible released oxygen to ameliorate the exothermic reaction with electrolyte; (4) the engineered LMO decorator would act as a physical barriers to improve the interfacial stability through alleviating possible side-reactions. This study is novel in proposing a scalable strategy that fundamentally diminish the structure degradation resulted by oxygen species evolutions, which could bring layered cathode working at a moderate condition to reach widespread commercial application and broaden the perspectives for achieving next-generation LIBs and other rechargeable batteries.

## Methods

### Materials synthesis

Pristine LiNi$_{0.8}$Co$_{0.1}$Mn$_{0.1}$O$_2$ (P-NCM) was obtained by mixing Ni$_{0.8}$Co$_{0.1}$Mn$_{0.1}$(OH)$_2$ precursor (Ronbay Technology) and LiOH·H$_2$O (Aladdin) compound with 1:1.05 in molar ratio, then calcined at 400 °C for 6 h (5 °C min$^{-1}$) and 750 °C for 15 h (1 °C min$^{-1}$) in oxygen atmosphere. For the preparation of La$_2$Mo$_2$O$_9$ (LMO) modified material, stoichiometric P-NCM, La$_2$O$_3$ (Aladdin), and MoO$_3$ (Aladdin) were dissolved in alcohol and stirred (800 r min$^{-1}$) thoroughly for 6 h, then the solvent was evaporated at 80 °C. Finally, the compound was

annealed at 500 °C for 12 h (5 °C min⁻¹) to obtain $La_2Mo_2O_9$ (LMO) modified $LiNi_{0.8}Co_{0.1}Mn_{0.1}O_2$ (L-NCM). The graphite (Gr) was purchased from Sigma−Aldrich. In the pretest, 3 wt.% LMO-modified sample showed superior capability, thus abbreviated as L-NCM for hereafter analysis.

## Physical characterizations

The crystal structure was monitored by X-ray diffraction (XRD, Bruker D8 Advance) with Cu-Kα radiation at a scan increment of 0.020°, wherein the sample was pressed and flatted on a high-purity quartz wafer, then the data was fitted by Rietveld refinement (FullProf software). For in situ XRD characterization, the slurry was first coated and dried on a beryllium window (for X-ray penetration), whereafter assembled into a Bruker specially made chamber to carry out the test with the increment of 0.020°. In situ heating XRD (HT-XRD) was carried out with the increment of 0.020° for the disassembled delithiated (4.5 V) electrodes with the temperature elevating from 25 to 600 °C. The chemical environment was detected by X-ray photoelectron spectroscopy (XPS, PHI 5000 Versa Probe-III) and calibrated with C 1$s$ posited at 284.8 eV. The argon-ion etching for Li metal anode on in-depth XPS characterizations were corresponding to 1 s -100 nm. The fine structure for the powder and cycled electrode (disassemble the coin-cell) were measured by soft X-ray absorption spectroscopy (s-XAS, Hefei, National Synchrotron Radiation Laboratory) in total electron yield mode (TEY, with a detection depth ~15 nm), postmortem electron paramagnetic resonance (EPR, Bruker A300-10) and high-energy X-ray absorption near edge structure (XANES, Shanghai, Synchrotron Radiation Facility). The standard s-XAS spectral for $Ni^{2+}$, $Ni^{3+}$, and $Ni^{4+}$ were obtained by detecting the NiO, $Ni_2O_3$, and $NiO_2$ samples. The cycled cathodes (disassemble the coin cell) surface chemical compositions were then identified by time-of-flight secondary ion mass spectroscopy (TOF-SIMS, PHI Nano Tof I). As for scanning electron microscopy (SEM) characterizations, the samples were stuck directly into conductive adhesive for testing. Whereas in transmission electron microscopy (TEM) detections, the materials were first dispersed in alcohol and then dropped into carbon-based porous film, the test could be proceeded until it was completely dried. The contents of residual lithium in Ni-rich cathode surface were measured and calculated by a neutralization titration method: a certain amount of Ni-rich material was first dissolved in ultrapure water, and the solution was precisely titrated with 0.02 M HCl on an automatic titration device using ultrapure water as the blank after fully stirring and filtering the suspension. Finite element analysis was performed on COMSOL Multiphysics 6.1.

## Electrochemical measurements

Cathodes were prepared by mixing 80 wt.% active material, 10 wt.% conductive carbon, and 10 wt.% PVDF dissolved into N-Methyl-2-pyrrolidinone (NMP). Afterward, the slurry was coated onto aluminum foil and dried at 90 °C (12 h) in vacuum box. CR2032-type half-cells were assembled in an argon-filled glove box (MIKROUNA) composed of a high loading cathode (thickness: ~80.0 μm, diameter: ~14.0 mm, mass loading: ~10.0 mg cm⁻²), 1 M $LiPF_6$ dissolved in ethylene/diethyl carbonate (EC/DEC, 140 μL, volume ratio: 3:7) and 3 wt.% fluoroethylene carbonate (FEC) as the electrolyte, Li-foil (diameter: ~15.0 mm) as the counter electrode and Celgaed-2400 film (diameter: ~16.2 mm) as the separator, respectively. The half-cells were tested in the voltage range of 2.7–4.3/2.7–4.4/2.7–4.5 at the current of C/3 (1 C = 220 mA g⁻¹). The L-NCM cathode and Gr anode were employed to assemble a pouch-type full cell. The mass loading for P-NCM and Gr was 20.00 mg cm⁻² and 12.48 mg cm⁻², respectively. Furthermore, the ratio of negative to positive electrode capacity (N/P ratio) was 1.12 ± 0.01. The separator and electrolyte were same as those applied in half-cells. The parameters for full-cell were also presented in Supplementary Table 6. The cycling voltage range was 2.8–4.2 V, and five-

formation cycles were conducted at 0.1 C before long-term cycling at 0.5 C. Specifically, the formation procedure of the full cell is first charging to 75% state of charge (SOC) in initial cycle, and then standing for 24 h at 30 °C. After that, charging the full cell to 100% SOC and then performed the discharge procedure. The galvanostatic charge and discharge tests were performed using a Land battery testing system. Electrochemical impedance spectra (EIS) and cyclic voltammetry (CV) tests were carried out on Neware electrochemical workstation. All the electrochemical tests were carried out at 30 °C.

## Theoretical methods

The density functional theory (DFT) calculation was performed using the Vienna ab initio Simulation Package (VASP 5.4.1) with the projector augmented-wave potentials[34,35]. The generalized gradient approximation (GGA) with the Perdew-Burke-Ernzerhof (PBE) functional was employed to describe the exchange-correlation[36,37]. A cut-off energy for the plane-wave basis was set to 520 eV for all calculations. The energy convergence criteria for self-consistent-field iteration were $10^{-5}$ eV, and the atomic positions were fully optimized until all the residual forces were smaller than 0.01 eV Å⁻¹. The $LiNiO_2$ (104, LNO) slab and LMO (001) slab models were first created by the pre-optimized bulk LNO and LMO, respectively, and a 15 Å vacuum space along the **z**-direction was included to avoid inter-layer interactions. The adsorption models of La and Mo atoms on the LNO (104) surface were then constructed, respectively, from the optimized LNO (104) slab to mimic the LMO deposited on LNO. Finally, the presence of single oxygen defect on both the LNO (104) surface and the LMO (001) surface were simulated to compare the oxygen vacancy deformation energy. For all the slab simulation, the DFT − D3 Grimme strategy for dispersion correction of total energy was used to include the vdw interaction[38].

Bader charge analysis was performed on the adsorption of La/Mo for the LNO surface. The O vacancy deformation energy was calculated by:

$$E_F = E_{slab} - E_{sub} + E_O \qquad (1)$$

wherein $E_{slab}$ and $E_{sub}$ are the total energy of $LiNiO_2$ (or LMO) slab with and without oxygen defect, respectively, and $E_O$ is the energy of single oxygen atom.

## Data availability

All relevant data that support the findings of this study are presented in the manuscript and supplementary information file. Source data are provided with this paper.

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

## Acknowledgements

This work was supported by the National Natural Science Foundation of China (52374410 to L.L.), Beijing Natural Science Foundation (Z220021 to L.L.), National Natural Science Foundation of China (52002024 to R.C.), National Natural Science Foundation of China (22202011 to Z.L.) and Shandong Provincial Natural Science Foundation (ZR2022QB056 to Z.L.).

## Author contributions

L.L., F.W., and R.C. proposed and supervised the project. Z.D. carried out the experiments and data analysis. Z.L. performed the theoretical calculations. All the authors discussed and commented on the manuscript.

## Competing interests

The authors declare no competing interests.
