## [Peer Review File · Nature Communications]

Defective oxygen inert phase stabilized high-voltage Ni-rich cathode for high-energy lithium-ion batteriesREVIEWER COMMENTS

Reviewer #1 (Remarks to the Author):

In this manuscript, the authors discuss the effect of stabilizing the oxygen-defective inert phase on the cathode properties of nickel-rich cathode material from various perspectives, such as bulk structure and surface structure, in addition to electrochemical properties. The discussion has many very interesting contents, but the reviewer had doubts about the following contents.

1. The Ni L-edge XAS spectrum of each cathode after 4.5 V charged state results are shown in Fig. 1 (h). The authors claim that Ni⁴⁺ is decreasing based on this result, but in addition to the initial results, I think that necessary information such as the shape of the Ni³⁺ spectrum is lacking.
2. Although the oxygen vacancies are discussed, it is presumed that the oxygen vacancies are affecting other electronic states of oxygen. The difference between P-NCM and L-NCM with EPR and XPS has been debated. If you measure O K-edge spectra using s-XAS, I think you can understand more details. You are measuring Ni L-edge spectra, but are you also measuring O results? DFT calculations have also been carried out, so I think we can understand the details further by discussing them in combination with the results.

Reviewer #2 (Remarks to the Author):

The article which I received to review from Dai et al. proposed a strategy of using electron-donating, oxygen-defect-rich La₂Mo₂O₉ layer as surficial protection on lithium metal oxide cathodes to suppress the lattice oxygen evolution and capture it, if generated. This strategy improves the high-voltage cathode stability, as the protection layer mitigates the lattice rearrangement, stress accumulation and particle fracturing due to oxygen release. The results demonstrated improved performance. Many advanced characterization techniques with theoretical simulations verified the improved stability from interface to internal. It is a great work with novel design and decent understanding. But there are many places that remain ambiguous and confusing, which need improvements. I think this work can be published in Nature Communications but after a major revision based on the comments below.

1. The authors observed improved stability from both interfacial and bulk characterizations from the L-NMC sample, while the material is only subjected to a surficial modification (LMO deposition). I think it will be meaningful if the authors can discuss and highlight how interfacial stability can propagate to a bulk structural stability.
2. Page 5 line 9, the authors need to provide references for XPS peak assignments. And if LMO is highly enriched with O vac., why does the L-NCM vacancy signal only marginally increase in supplementary figure 6. As depth of XPS detection is several nm, the signal of L-NCM should largely originate from LMO. Also, why there is no lattice oxygen from P-NCM.
3. Page 5 line 19, is the O release due to receiving electron from transition metal or losing electron to it? And I guess the authors might meant to say “band” instead of “bond” there.
4. Figure 1c EDS, why do the La and Mo signals completely overlap with Ni-Co-Mn, if the LMO is surficial deposition?
5. There is overall very little information on sample preparations for nearly all the characterizations.

6. The coulombic efficiency must be included in Fig 2g-i as they are direct indicatives of the cathode side reaction extents, especially at high voltage.
7. Supplementary Fig 18. The LiF from L-NCM decreased, but the other F-containing compound largely increased. How did the authors conclude mitigated electrolyte decomposition from it? And why is there a large increase in the OCO₂ signal from P-NCM?
8. Supplementary Fig 19. It's very difficult to conclude there is noticeable decreasing at all, upon etching.
9. The SEM Fig 4 g, h and the discussions are very misleading. First, it's very unlikely that any SEI can be observed at such magnification. The SEM images here can only show the Li deposition morphology. The top-view even needs zoom-in for just observing detailed Li morphology. Second, the LMA cycled in the carbonate electrolyte that the authors use in this work is highly impossible to deliver smooth deposition, even without any interference from cathode. The 4h-right figure does not indicate the SEI information. Instead, the observed layer is the porous Li deposition after long-term LMA cycling. Therefore, the smooth surface of 4h-left does not make sense. Here I do not suspect that mitigating the TM crosstalk can improve the Li deposition. But better images are needed with correct interpretations.
10. What's the depth of detection from XANES and EXAFS used in this work? Are they suitable for detecting surficial changes without being affected by bulk information?
11. Figure 5i. Where is the LMO deposition layer which is supposed to be on the NCM surface with the thickness at around 5 nm as indicated in Fig 1c?

Reviewer #3 (Remarks to the Author):

- 1) The manuscript reports an importance class of cathode materials in the field of Lithium-ion battery, with detailed and very insightful experimental and computational investigations. The results show clearly that the use of lanthanum molybdate improves the properties of P-NCM. However, it is necessary to compare these results with those similar-focused studies reported in the literature, especially experimental results, in order to better appreciate the importance of the present study. The following literature can help you:
<https://doi.org/10.1016/j.ensm.2021.06.018>, <https://doi.org/10.1038/s41560-021-00780-2>,
<https://doi.org/10.1016/j.mattod.2020.01.019>
- 2) A relationship must be established between experimental and theoretical results. Do the theoretical results agree with the experimental ones? If yes, specify in the manuscript.
- 3) Please the quality of your image should be improved.
- 4) The slight challenges I encountered during the review has to do with the language. The English of the manuscript needs extensive revision.

Detailed Responses to Reviewers' Comments

Reviewer #1:

General comment: In this Manuscript, the authors discuss the effect of stabilizing the oxygen-defective inert phase on the cathode properties of nickel-rich cathode material from various perspectives, such as bulk structure and surface structure, in addition to electrochemical properties. The discussion has many very interesting contents, but the reviewer had doubts about the following contents.

Reply: Thanks for your valuable advice and professional comment to help us improving the manuscript. Your suggestions and comments have been carefully considered and addressed, and all the responses are listed in the following section, point by point. In addition, all changes in the revised Manuscript have been highlighted in yellow for your convenience.

1. The Ni L-edge XAS spectrum of each cathode after 4.5 V charged state results are shown in Fig. 1 (h). The authors claim that Ni⁴⁺ is decreasing based on this result, but in addition to the initial results, I think that necessary information such as the shape of the Ni³⁺ spectrum is lacking.

Reply: Thanks for your professional comment. As reported in previous literatures (*e.g.*, Doeff *et al. Chem. Rev.* 2017, 117, 13123; Xin *et al. Nature* 2022, 6, 610), the information of Ni²⁺, Ni³⁺ and Ni⁴⁺ in Ni-rich cathode surface could be obtained by comparing the Ni L₃-edge s-XAS spectra (TEY mode) with their standard shape. Therefore, we detected the NiO, Ni₂O₃ and NiO₂ samples to obtain the standard spectral of Ni²⁺, Ni³⁺ and Ni⁴⁺, and then diagramed the results in Ni L₃-edge s-XAS profiles to illustrate the information of Ni ions at various valences. As shown in the following revised Fig. 1h, the relative contents of Ni²⁺, Ni³⁺ and Ni⁴⁺ ions were unambiguously elucidated. Furthermore, according to your suggestion, the Ni L₃-edge s-XAS spectrum of each uncharged cathode (initial state) have been monitored and discussed. (added as Supplementary Fig. 10 in the revised *Supplementary Information*). To clarify more clearly, the more detailed explanation of Ni-L₃-edge s-XAS spectra have been strengthened in the revised Manuscript and *Supplementary Information*.

Origin 1: (On page 6 in the original Manuscript)

Revision 1: (On page 6 in the revised Manuscript)

Origin 2: (On page 7 in the original Manuscript)

To elucidate the alleviated electrons transfer initiated from TMs and surface oxygens, the high surface-sensitive soft X-ray absorption spectroscopy (s-XAS) of Ni L_3 -edge at highly delithiated state (4.5 V, Fig. 1h) was detected and indicated that the Ni^{4+} was decreased in P-NCM sample near-surface, manifesting that the bond gap in L-NCM might be regulated¹⁶.

Revision 2: (On page 7 in the revised Manuscript)

To elucidate the alleviated electron transfer between TMs and surface oxygens, the s-XAS of Ni L_3 -edge with TEY mode at 1st fully charged state (4.5 V, Fig. 1h) were detected. Based on the standard spectral of NiO, Ni₂O₃ and NiO₂, the Ni^{3+} and Ni^{4+} were decreased in P-NCM sample near-surface, whereas the content of Ni^{2+} was elevated compared with their counterpart, manifesting that the charge transfer in L-NCM near-surface were mitigated¹⁶.

Addition 1: (On page 11 in the revised *Supplementary Information*)

Supplementary Fig. 10. s-XAS spectra for Ni L_3 -edge at uncharged state.

Addition 2: (On page 5 in the revised Manuscript)

Additionally, the slightly decreased Ni^{2+} near L-NCM surface (Supplementary Fig. 10) could induce the mitigated Li^+/Ni^{2+} cations disorder, consistent with the XPS results.

Addition 3: (On page 22 in the revised Manuscript, in the Physical characterizations section)

The standard s-XAS spectral line for Ni^{2+} , Ni^{3+} and Ni^{4+} were obtained by detecting the NiO, Ni_2O_3 and NiO_2 samples.

2. Although the oxygen vacancies are discussed, it is presumed that the oxygen vacancies are affecting other electronic states of oxygen. The difference between P-NCM and L-NCM with EPR and XPS has been debated. If you measure O K-edge spectra using s-XAS, I think you can understand more details. You are measuring Ni L-edge spectra, but are you also measuring O results? DFT calculations have also been carried out, so I think we can understand the details further by discussing them in combination with the results.

Reply: Thanks for your valuable comment. As the reviewer suggested, the O K-edge s-XAS spectra for the uncharged, 1st charged cycle and 200th charged cycle samples were measured. Combined with the XPS, EPR, in-situ DEMS and DFT calculations results, more detailed discussion and comparison have been strengthened in the revised Manuscript.

Addition 1: (On page 10 in the revised *Supplementary Information*)

Supplementary Fig. 9. s-XAS spectra for O *K*-edge at uncharged state.

Addition 2: (On page 5 in the revised Manuscript)

The high surface-sensitive O *K*-edge X-ray absorption spectroscopy (s-XAS) with total electron yield (TEY, with a detection depth ~ 15 nm) mode could provide the information of oxygen *2p* orbital information¹⁶. As shown in Supplementary Fig. 9, the enhanced pre-edge (~ 529 eV) in uncharged L-NCM corresponding to the decreased charge density around the O ions and illustrating more oxygen vacancies¹⁷, which could be attributed to the oxygen defect in LMO layer. Furthermore, the high relative intensity of L-NCM electron paramagnetic resonance (EPR) spectroscopy (Fig. 1e) confirmed that the total oxygen vacancies were higher than their pristine counterpart. Additionally, slightly decreased Ni^{2+} near L-NCM surface (Supplementary Fig. 10) could induce the mitigated $\text{Li}^+/\text{Ni}^{2+}$ cations disorder, consistent with the XPS results.

Addition 3: (On page 25 in the revised Manuscript)

16 Lin, F. et al. Synchrotron X-ray analytical techniques for studying materials electrochemistry in rechargeable batteries. *Chem. Rev.* **117**, 13123 (2017).

17 Lin, F. et al. Phase evolution for conversion reaction electrodes in lithium-ion batteries. *Nat. Commun.* **5**, 3358 (2014).

Addition 4: (On page 12 in the revised *Supplementary Information*)

Supplementary Fig. 11. a s-XAS spectra for O K -edge at 1st charged state. **b** EPR profile at 1st charged state.

Addition 5: (On page 7 in the revised Manuscript)

The O K -edge s-XAS spectra for the 1st charged state (Supplementary Fig. 11a) exhibited that the TM 3d-oxygen 2p rehybridization in P-NCM was elevated, whereas the pre-edge in L-NCM was almost unchanged, demonstrating that the more oxygen vacancies were generated in P-NCM near-surface and the mitigated oxygen activities in L-NCM, coinciding with the in-DEMS analysis and could be testified by the EPR tests (Supplementary Fig. 11b).

Addition 6: (On page 13 in the revised *Supplementary Information*)

Supplementary Fig. 13. a s-XAS spectra for O K -edge at 200th charged state. **b** EPR profile at 200th charged state.

Addition 7: (On page 7 in the revised Manuscript)

Furthermore, the severely enhanced O s-XAS pre-edge and the relative intensity of EPR profile for 200th charged P-NCM sample illustrated the abundant oxygen vacancies (Supplementary Fig. 13). Nevertheless, the intensities in L-NCM were slightly enhanced compared with the uncharged state, confirmed the decreased oxygen activities after LMO modification.

Reviewer #2:

General comment: The article which I received to review from Dai et al. proposed a strategy of using electron-donating, oxygen-defect-rich $\text{La}_2\text{Mo}_2\text{O}_9$ layer as surficial protection on lithium metal oxide cathodes to suppress the lattice oxygen evolution and capture it, if generated. This strategy improves the high-voltage cathode stability, as the protection layer mitigates the lattice rearrangement, stress accumulation and particle fracturing due to oxygen release. The results demonstrated improved performance. Many advanced characterization techniques with theoretical simulations verified the improved stability from interface to internal. It is a great work with novel design and decent understanding. But there are many places that remain ambiguous and confusing, which need improvements. I think this work can be published in Nature Communications but after a major revision based on the comments below.

Reply: Thanks for your valuable advice and professional comment to help us improving the manuscript. Your suggestions and comments have been carefully considered and addressed, and all the responses are listed in the following section, point by point. In addition, all changes in the revised Manuscript have been highlighted in yellow for your convenience.

1. The authors observed improved stability from both interfacial and bulk characterizations from the L-NMC sample, while the material is only subjected to a surficial modification (LMO deposition). I think it will be meaningful if the authors can discuss and highlight how interfacial stability can propagate to a bulk structural stability.

Reply: Thanks for your professional suggestion. Actually, the performance degradation of Ni-rich cathode was caused by the cascade reaction of various failure mechanisms. Among them, the c -axis collapse at high voltage have been believed to result in severely bulk structural instability. As widely reported in previous literatures (*e.g.*, Peter *et al. Nat. Chem.* 2016, 8, 684; Meng *et al. Nat. Rev. Mater.* 2022, 7, 522), the shrinkage of lattice c in layered materials at highly delithiated state could be attributed to the rehybridization of high valence transition metal ions (TMs, particularly for Ni^{4+} in Ni-rich material) with oxygen. In this process, the high valence and strong oxidizing TMs would receive electrons from the coordinate oxygen, promoting the oxygen p orbitals approaching to the Fermi level E_f , which leading to the TMs $3d$ -oxygen $2p$ orbital overlap and

thus the oxygen instability. The oxygen release in layered material would decrease the coulomb repulsion of oxygen plane in LiO layer, thus resulting in the *c*-axis shrinkage, and it was generally believed that the oxygen release was initiated from surface and extended to the bulk (e.g., Sun *et al. Adv. Energy Mater.* 2022, 2200569; Wang *et al. Nat. Commun.* 2018, 9, 2437). Therefore, the suppression of oxygen release from the surface could effectively improve the bulk stability of Ni-rich material.

In this work, the enhanced bulk structural stability of L-NCM could be ascribed to the strong La-O and Mo-O formed in the interface. The La and Mo ions would spontaneously transfer electrons to the coordinate oxygen, reducing its electronic states near the Fermi level E_f and improving the intrinsic stability, which has been proved by the DFT simulations (Fig. 6a-f). Additionally, the H2-H3 redox peak (Fig. 2b) and *c*-axis (Fig. 3c) shrinkage of L-NCM was postponed to a higher voltage compared with P-NCM, which is a direct evidence that the interfacial electrons transfer has been involved and thus the decreased TMs 3*d*-oxygen 2*p* orbital overlap. Therefore, the LMO layer could not only inhibited the interface failure caused by side reactions through the physical barrier effect, but also weakened the change of lattice *c* through the electron transport between it with the bulk Ni-rich material, thus improving the bulk stability. For clearly, the corresponding description has been strengthened in the revised Manuscript.

Addition 1: (On page 12 in the revised Manuscript)

The enhanced structural stability of L-NCM might be ascribed to the surface electrons transition from La and Mo ions to the coordinate oxygen, elevating the oxygen electronegativities and thus mitigating TMs 3*d*-oxygen 2*p* orbital overlap at highly delithiated state, eventually alleviated the lattice collapse resulted by the oxygen release from surface to bulk.

2. Page 5 line 9, the authors need to provide references for XPS peak assignments. And if LMO is highly enriched with O vac., why does the L-NCM vacancy signal only marginally increase in supplementary figure 6. As depth of XPS detection is several nm, the signal of L-NCM should largely originate from LMO. Also, why there is no lattice oxygen from P-NCM.

Reply 1: Thanks for your constructive suggestion, As the reviewer suggested, relevant references for XPS peak assignments have been provided in the revised Manuscript. Furthermore, detailed descriptions for the XPS peak assignments have been strengthened in the revised *Supplementary*

Information.

Addition 1: (On page X in the revised Manuscript)

14 Wang, Q. et al. Unlocking anionic redox activity in O3-type sodium 3d layered oxides via Li substitution. *Nat. Mater.* **20**, 353-361 (2021).

Addition 2: (On page 8 in the revised *Supplementary Information*)

As shown in the fitted XPS spectra, the signals of La and Mo elements appeared in L-NCM surface (Supplementary Fig. 7a, b), which could be assigned to the 3d spin-orbit doublet^{1,2}, demonstrating the successful LMO deposition. Furthermore, the ratio of O vacancy located at 532.2 eV was obviously elevated after LMO modification (Supplementary Fig. 7c)³, confirming an enriched oxygen vacancy interface were established in L-NCM near-surface. Moreover, the Ni 2p signals could be divided into two peaks at 854.7, and 856.5 eV (the purple is satellite peak, Supplementary Fig. 7d)⁴, which were ascribed to 2p_{3/2} of Ni²⁺ and Ni³⁺ cations⁴, respectively.

Addition 3: (On page X in the revised *Supplementary Information*)

1 Zhu, X. et al. Epitaxial growth of an atom-thin layer on a LiNi_{0.5}Mn_{1.5}O₄ cathode for stable Li-ion battery cycling. *Nat. Commun.* **13**, 1565 (2022).

2 Geon, P. et al. Introducing high-valence elements into cobalt-free layered cathodes for practical lithium-ion batteries. *Nat. Energy* **7**, 946-954 (2022).

3 Wang, Q. et al. Unlocking anionic redox activity in O3-type sodium 3d layered oxides via Li substitution. *Nat. Mater.* **20**, 353-361 (2021).

4 Dai, Z. et al. Surface coupling between mechanical and electric fields empowering Ni-rich cathodes with superior cyclabilities for lithium-ion batteries. *Adv. Sci.* **9**, e2200622 (2022).

Reply 2: Thanks for your valuable comment. We chose a more suitable baseline to refit the XPS peak and the results shown in the revised Supplementary Fig. 7c, the obviously enhanced oxygen vacancy could be observed in L-NCM surface, which could be attributed to the LMO deposition (enriched oxygen vacancy) as the depth of XPS detection in this work is ~ 5 nm (detection depth for this device are 5-10 nm). Furthermore, the O K-edge s-XAS (detection depth: ~ 15 nm) and EPR (could reflect the total oxygen vacancy) spectra for the uncharged, 1st charged cycle and 200th charged cycle samples were measured. Combined with the XPS, EPR and in-situ DEMS results, more detailed discussion and comparison of oxygen vacancies in L-NCM and P-NCM have been

strengthened in the revised Manuscript.

Origin 1: (On page 7 in the original *Supplementary Information*)

Revision 1: (On page 8 in the revised *Supplementary Information*)

Addition 1: (On page 5 in the revised Manuscript)

which might originate from the LMO (enriched oxygen defect) layer.

Addition 2: (On page 10 in the revised *Supplementary Information*)

Supplementary Fig. 9. s-XAS spectra for O K-edge at uncharged state.

Addition 3: (On page 5 in the revised Manuscript)

The high surface-sensitive O K-edge X-ray absorption spectroscopy (s-XAS) with total electron

yield (TEY, with a detection depth ~ 15 nm) mode could provide the information of oxygen $2p$ orbital information¹⁶. As shown in Supplementary Fig. 9, the enhanced pre-edge (~ 529 eV) in uncharged L-NCM corresponding to the decreased charge density around the O ions and illustrate more oxygen vacancies¹⁷, which could be attributed to the intrinsic oxygen defect in LMO layer. Furthermore, the high relative intensity of L-NCM electron paramagnetic resonance (EPR) spectroscopy (Fig. 1e) confirmed that the total oxygen vacancies were higher than their pristine counterpart. Additionally, slightly decreased Ni^{2+} near L-NCM surface (Supplementary Fig. 10) could induce the mitigated $\text{Li}^+/\text{Ni}^{2+}$ cations disorder, consistent with the XPS results.

Addition 4: (On page X in the revised Manuscript)

16 Lin, F. et al. Synchrotron X-ray analytical techniques for studying materials electrochemistry in rechargeable batteries. *Chem. Rev.* **117**, 13123 (2017).

17 Lin, F. et al. Phase evolution for conversion reaction electrodes in lithium-ion batteries. *Nat. Commun.* **5**, 3358 (2014).

Addition 5: (On page 12 in the revised *Supplementary Information*)

Supplementary Fig. 11. **a** s-XAS spectra for O K -edge at 1st charged state. **b** EPR profile at 1st charged state.

Addition 6: (On page 7 in the revised Manuscript)

The O K -edge s-XAS spectra for the 1st charged state (Supplementary Fig. 11a) exhibited that the TM $3d$ -oxygen $2p$ rehybridization in P-NCM was elevated, whereas the pre-edge in L-NCM was almost unchanged, demonstrating that the more oxygen vacancies were generated in P-NCM near-surface and the mitigated oxygen activities in L-NCM, coinciding with the in-DEMS analysis and could be testified by the EPR tests (Supplementary Fig. 11b).

Addition 7: (On page 13 in the revised *Supplementary Information*)

Supplementary Fig. 13. a s-XAS spectra for O *K*-edge at 200th charged state. **b** EPR profile at 200th charged state.

Addition 8: (On page 7 in the revised Manuscript)

Furthermore, the severely enhanced O s-XAS pre-edge and the relative intensity of EPR profile for 200th charged P-NCM sample illustrated the abundant oxygen vacancies (Supplementary Fig. 13). Nevertheless, the intensities in L-NCM were slightly enhanced compared with the uncharged state, confirmed the decreased oxygen activities after LMO modification.

Reply 3: In the refitted XPS spectra, the surface oxygen (Li_2CO_3 /LiOH) in P-NCM surface was elevated, whereas the lattice oxygen was evidently mitigated compared with the modified sample. In this detection depth (~ 5 nm), the more relative content of surface oxygen in P-NCM was ascribed to the intrinsic residual lithium (Cho *et al. Adv. Mater.* 2018, 5, 30) in Ni-rich material, and the decreased content in L-NCM surface could be attributed to the alcohol cleaning effect during the modification process (Arumugam *et al. ACS Energy Lett.* 2021, 6, 941). To confirm the alleviated residual lithium species in L-NCM, a well-designed neutralization titration experiment has been carried out, the results unambiguously exhibited the decreased residual lithium in L-NCM sample. For clearly, the corresponding results and descriptions have been strengthened in the modified Manuscript and *Supplementary Information*.

Addition 1: (On page 9 in the revised *Supplementary Information*)

Supplementary Fig. 8. Residual lithium content in P-NCM and L-NCM.

Addition 2: (On page 5 in the revised Manuscript)

The decreased surface oxygen would be effectively alleviate the slurry flocculation during the mixing process¹⁵, which could be manifested by the residual lithium content detect (Supplementary Fig. 8).

15 Kim, Y. et al. Unraveling the intricacies of residual lithium in high-Ni cathodes for lithium-ion batteries. *ACS Energy Lett.* **6**, 941 (2021).

Addition 3: (On page 22 in the revised Manuscript)

The contents of residual lithium in Ni-rich cathode surface were measured by a neutralization titration method: a certain amount of the Ni-rich powder was first dissolved in ultrapure water, after fully stirred and filtered the suspension, the solution was precisely titrated with 0.02 M HCl on an automatic titration device using ultrapure water as the blank. The contents of residual lithium were then calculated from the neutralization reaction.

3. Page 5 line 19, is the O release due to receiving electron from transition metal or losing electron to it? And I guess the authors might meant to say “band” instead of “bond” there.

Reply 1: Thanks for your valuable comment. As established in previous literatures (*e.g.*, Xin *et al. Nat. Mater.* 2023, 22, 235; Ouyang *et al. Nat. Energy.* 2022, 7, 808), the elevated TMs valence in Ni-rich cathodes would lead to the decrease of 3d energy band upon charging, and when it contacted with the top of oxygen 2p band, the highly oxidized TMs (Ni⁴⁺ in particular) would receive electrons from oxygen, which resulted in the oxygen release and phase transition (Ni migrated to Li layer). Therefore, the “transport to” modified as “transport from” is more

appropriate, and all the inappropriate corresponding description have been corrected in the revised Manuscript. Furthermore, corresponding literatures have been added to strengthen the description.

Origin 1: (On page 5 in the original Manuscript)

resulting in the electrons transport to oxygen and thus the gas liberation would be happened.

Revision 1: (On page 5 in the revised Manuscript)

resulting in the electrons **transport from** oxygen and thus the gas liberation would be happened.

Origin 2: (On page 25 in the original Manuscript)

was primarily ascribed to the electrons transferred from high valence TMs with the coordinated oxygen.

Revision 2: (On page 25 in the revised Manuscript)

was primarily ascribed to the high valence TMs **received electrons from** the coordinated oxygen^{19,20}.

Addition: (On page X in the revised Manuscript)

19 Liu, X. et al. Origin and regulation of oxygen redox instability in high-voltage battery cathodes. *Nat. Energy* 7, 808-818 (2022).

20 Wang, C. et al. Resolving complex intralayer transition motifs in high-Ni-content layered cathode materials for lithium-ion batteries. *Nat. Mater.* 22, 235-241 (2023).

Reply 2: Thanks for your professional reminding. As the reviewer suggested, the “bond” there have been revised as “band”. Furthermore, all the inappropriate corresponding description have been corrected in the revised Manuscript.

Origin 1: (On page 5 in the original Manuscript)

the bottom of TMs 3d bond might be contact with the oxygen 2p bond.

Revision 1: (On page 5 in the revised Manuscript)

the bottom of TMs 3d **band** might be contact with the oxygen 2p **band**.

Origin 2: (On page 2 in the original Manuscript)

when it contacts with the bond of lattice oxygen.

Revision 2: (On page 2 in the revised Manuscript)

when it contacts with the **band** of lattice oxygen.

4. Figure 1c EDS, why do the La and Mo signals completely overlap with Ni-Co-Mn, if the LMO

is surficial deposition?

Reply: Thanks for your meticulous comment. The EDS mapping images in Fig. 1c were obtained by the energy spectrum of surface distribution, thus the uniformed La and Mo signals has been detected in the surface (overlap with Ni-Co-Mn). To demonstrate the LMO compound were only deposited on L-NCM surface, a focused ion beam etching combined with scanning electron microscope (FIB-SEM) technique was employed. The cross-sectional mapping images shown in the following Fig. (added as Supplementary Fig. 6 in the revised *Supplementary Information*) unambiguously demonstrated that the LMO is surficial deposition. To clarify more clearly, the corresponding result and explanation has been strengthened in the revised Manuscript *and Supplementary Information*.

Addition 1: (On page 4 in the revised Manuscript)

Specially, cross-section morphology of L-NCM obtained by focused ion beam etching combined with SEM (FIB-SEM) and the corresponding energy dispersive spectra (EDS) mapping images (Supplementary Fig. 6) demonstrated that La and Mo elements were uniformly distributed on the materials surface, illuminating the homogeneous LMO surface deposition.

Addition 2: (On page 7 in the revised *Supplementary Information*)

Supplementary Fig. 6. Cross-section morphology and the corresponding EDS mapping images for L-NCM, scale bars, 3 μm .

5. There is overall very little information on sample preparations for nearly all the

characterizations.

Reply: Thanks for your kind reminding. As the reviewer recommend, the detailed sample preparations for all the characterizations have been strengthened in the revised Manuscript.

Origin 1: (On page 22 in the original Manuscript)

Crystal structural was monitored by X-ray diffraction (XRD, Bruker D8 Advance) and quantified by Rietveld refinement (Fullprof soft-ware).

Revision 1: (On page 22 in the revised Manuscript)

Crystal structural was monitored by X-ray diffraction (XRD, Bruker D8 Advance) with Cu-K α radiation at a scan increment of 0.020°, wherein the sample were pressed and flat on a high-purity quartz wafer, then the crystal structural was quantified by Rietveld refinement (Fullprof soft-ware).

Origin 2: (On page 22 in the original Manuscript)

In-situ XRD test was performed in the scanning range of 10-55° with Be window.

Revision 2: (On page 22 in the revised Manuscript)

For in-situ XRD characterization, the slurry was firstly coated and dried on a beryllium window (for X-ray penetration), whereafter assembled into a Bruker specially made chamber to carried out the test with the increment of 0.020° in the scanning range of 10-55°.

Origin 3: (On page 22 in the original Manuscript)

In-situ heating XRD (HT-XRD) was carried out on delithiated electrodes with the temperature elevated from 25 to 600 °C.

Revision 3: (On page 22 in the revised Manuscript)

In-situ heating XRD (HT-XRD) was carried out with the increment of 0.020° in the scanning range of 10-70° for the disassembled delithiated (4.5 V) electrodes with the temperature elevated from 25 to 600 °C.

Origin 4: (On page 22 in the original Manuscript)

Chemical environments were detected by X-ray photoelectron spectroscopy (XPS, PHI 5000 Versa Probe-III), X-ray absorption near edge structure (XANES, Shanghai Synchrotron Radiation Facility), soft X-ray absorption spectroscopy (s-XAS, Hefei, National synchrotron radiation Laboratory), postmortem electron paramagnetic resonance (EPR, Bruker A300-10), and time-of-flight secondary ion mass spectroscopy (TOF-SIMS, PHI Nano ToF I).

Revision 4: (On page 22 in the original Manuscript)

Chemical environments were detected by X-ray photoelectron spectroscopy (XPS, PHI 5000 Versa Probe-III) and calibrated with C 1s posited at 284.8 eV. The argon-ion etching for Li metal anode in-depth XPS characterizations were corresponding to 1 s ~ 100 nm. The fine structure for the original powers and cycled electrodes (disassemble the coin-cell) were measured by soft X-ray absorption spectroscopy (s-XAS, Hefei, National Synchrotron Radiation Laboratory) in total electron yield mode (TEY, with a detection depth ~ 15 nm), postmortem electron paramagnetic resonance (EPR, Bruker A300-10), and high-energy X-ray absorption near edge structure (XANES, Shanghai, Synchrotron Radiation Facility). The cycled cathodes (disassemble the coin-cell) surface chemical compositions were then identified by time-of-flight secondary ion mass spectroscopy (TOF-SIMS, PHI Nano ToF I).

Addition 1: (On page 22 in the revised Manuscript)

focused ion beam etching combined with SEM technique (FIB-SEM)

Addition 2: (On page 22 in the revised Manuscript)

As for scanning electron microscopy (SEM) characterizations, the samples were stick directly to conductive adhesive for testing. Whereas in transmission electron microscopy (TEM) detections, the materials were firstly dispersed in alcohol and then drop into carbon-based porous film, the tests could be proceeded until its completely dried.

6. The coulombic efficiency must be included in Fig. 2g-i as they are direct indicatives of the cathode side reaction extents, especially at high voltage.

Reply: Thanks for your constructive suggestion. As the reviewer suggested, the coulombic efficiency has been included in the revised Fig. 2g-i.

Origin: (On page 8 in the original Manuscript)

Revision: (On page 9 in the revised Manuscript)

7. Supplementary Fig 18. The LiF from L-NCM decreased, but the other F-containing compound largely increased. How did the authors conclude mitigated electrolyte decomposition from it? And why is there a large increase in the OCO₂ signal from P-NCM?

Reply 1: Thanks for your careful review. After referring the literatures (*e.g.*, Jiang *et al. Nat. Commun.* 2021, 12, 4564; Lucht *et al. Energy Environ. Sci.* 2016, 9, 1308), the F-contained XPS spectra have been re-fitted and shown in the following revised Supplementary Fig. 18b. Actually, the red peak (687.1 eV) was more properly assigned to C-F bonds and could be ascribed to the PVDF binder deposited on electrode surface, thus their relative amount for analyze the electrolyte decomposition could be ignored. The green (685.2 eV) and purple (683.8 eV) peaks were index to the Li_xPO_yF_z/Li_xPF_y and LiF, respectively, which were derived from the parasitic reactions at the electrode/electrolyte interface and were widely identified as the components of CEI films. Therefore, the almost disappeared Li_xPO_yF_z/Li_xPF_y and LiF peaks in L-NCM surface compared with P-NCM could be demonstrated the mitigated electrolyte decomposition.

As discussed and analyzed in the preceding text, relevant clearly description and graphs have been accordingly added and revised in the revised *Supplementary Information*.

Origin: (On page 19 in the *Supplementary Information*)

Revision: (On page 24 in the revised *Supplementary Information*)

Addition 1: (On page 24 in the *Supplementary Information*)

Additionally, as shown in Supplementary Fig. 23b, the LiF (684.3 eV) and LixPOyFz/LixPFy (685.2 eV) in F 1s peaks were derived from the parasitic reactions at the electrode/electrolyte interface and were identified as the components of cathode-electrolyte interface (CEI)^{7,8}. The C-F (687.1 eV) bonds could be ascribed to the PVDF binder deposited on electrode surface. Therefore, the almost disappeared LixPOyFz/LixPFy and LiF peaks in L-NCM surface compared with P-NCM demonstrated the mitigated electrolyte decomposition, coinciding with above-mentioned C 1s peaks analysis.

Addition 2: (On page 37 in the *Supplementary Information*)

7 Yu, H. et al. Surface enrichment and diffusion enabling gradient-doping and coating of Ni-rich cathode toward Li-ion batteries. *Nat. Commun.* **12**, 4564 (2021).

8 Xu, M. et al. Development of novel lithium borate additives for designed surface modification of high voltage LiNi_{10.5}Mn_{1.5}O₄ cathodes. *Energy Environ. Sci.* **9**, 1308 (2016).

Reply 2: The OCO₂ could be also denoted as CO₃, according to the in-depth analysis for the

decomposition mechanism of electrolyte in Ni-rich cathode surface carried by the scientists and established in recent literatures (e.g., Ming *et al. Adv. Mater.* 2021, 33, 2102964; Gery *et al. Energy Environ. Sci.* 2022, 5, 3416), the OCO₂ was also attributed to the electrolyte decomposition on Ni-rich electrode surface. Therefore, the large increase in the OCO₂ signal in P-NCM surface confirmed the more severely electrolyte decomposition. To clarify more clearly, the corresponding explanation has been complemented and strengthened in the revised *Supplementary Information*.

Addition 3: (On page 24 in the *Supplementary Information*)

As exhibited in C 1s peaks (Supplementary Fig. 23a), the C-C peaks at 285.1 eV originated the Super P, while the peaks located at 286.2 eV (C-H) were related to PVDF binder⁵. The C-O (287.2 eV), C=O (289.1 eV) and OCO₂ (290.2 eV) species were attributed from the electrolyte decomposition⁶, thus the increased C-O, C=O and OCO₂ intensity illustrated the aggravated electrolyte decomposition on P-NCM electrode surface.

Addition 4: (On page 37 in the *Supplementary Information*)

5 Bernardine, R. et al. Two electrolyte decomposition pathways at nickel-rich cathode surfaces in lithium-ion batteries. *Energy Environ. Sci.* **15**, 3416 (2022).

6 Zou, Y. et al. Interfacial model deciphering high-voltage electrolytes for high energy density, high safety, and fast-charging lithium-ion batteries. *Adv. Mater.* **33**, 2102964 (2021).

8. Supplementary Fig 19. It's very difficult to conclude there is noticeable decreasing at all, upon etching.

Reply: Thanks for your valuable reminding. As the reviewer suggested, a noticeable in-depth XPS profiles with mitigated Ni signals upon etching have been provided in the revised Supplementary Fig. 25. The results unambiguously proved that the Ni signals for both samples were decreased with increasing the etching times. Clearly, the more obvious weakened Ni peak intensity with etching in LMA surface (coupled with L-NCM) illuminated the alleviated TMs decomposition after LMO deposition.

Origin: (On page 19 in the original *Supplementary Information*)

Revision: (On page 25 in the revised *Supplementary Information*)

9. The SEM Fig. 4 g, h and the discussions are very misleading. First, it's very unlikely that any SEI can be observed at such magnification. The SEM images here can only show the Li deposition morphology. The top-view even needs zoom-in for just observing detailed Li morphology. Second, the LMA cycled in the carbonate electrolyte that the authors use in this work is highly impossible to deliver smooth deposition, even without any interference from cathode. The 4h-right figure does not indicate the SEI information. Instead, the observed layer is the porous Li deposition after long-term LMA cycling. Therefore, the smooth surface of 4h-left does not make sense. Here I do not suspect that mitigating the TM crosstalk can improve the Li deposition. But better images are needed with correct interpretations.

Reply 1: Thanks for your very professional comment. As the reviewer suggested, the SEM images with appropriate magnifications have been provided to clearly identify the SEI layer. Furthermore, the top-view has been zoomed-in and the results were displayed as an inset in the Top-view images. The results unambiguously revealed the detailed Li morphology.

Original: (On page 12 in the revised Manuscript)

g, h Top-view, and cross-sectional morphology images of LMA, scale bars, 100 μm. **(g)** coupled with P-NCM and **(h)** coupled with L-NCM.

Revision: (On page 13 in the revised Manuscript)

g Top-view (left, scale bars, 20 μm) and cross-sectional morphology (right, scale bars, 5 μm) images of LMA coupled with P-NCM. **h** Top-view (left, scale bars, 20 μm) and cross-sectional morphology (right, scale bars, 5 μm) images of LMA coupled with L-NCM.

Reply 2: Thanks for your very careful reviewing. The electrolyte in this work was purchased by Shandong Hirong Power Supply Material Co., LTD. As the reviewer mentioned that the carbonate electrolyte (ethylene/diethyl carbonate=3:7 in volume ratio) in this work is impossible to deliver smooth deposition, we reconfirmed that the electrolyte contained 3 wt% fluoroethylene carbonate (FEC) as an additive with the manufacturer. Therefore, the uniformed SEI layer in LMA surface could be attributed that the FEC compound facilitated the homogeneous deposition of Li metal (Zhang *et al. Angew. Chem. Int. Ed.* 2020, 59, 3252, **Fig. 2**). As established in the previous

literature (Cho *et al. Nat. Energy* 2021, 6, 362, **Fig. 5**), the dissolved TMs from Ni-rich cathode could deposit on the LMA surface, which would impede the Li deposition and affect the morphology and electrochemical performance. Therefore, the smooth LMA surface for those paired with L-NCM demonstrated that the LMO depositor has been effectively mitigated the TMs dissolution, corresponding to the excellent electrochemical performances.

Addition 1: (On page 22 in the revised Manuscript)

and 3 wt% fluoroethylene carbonate (FEC)

Addition 2: (On page 16 and 24 in the revised Manuscript)

3 Yoon, M. et al. Reactive boride infusion stabilizes Ni-rich cathodes for lithium-ion batteries. *Nat. Energy* 6, 362-371 (2021).

10. What's the depth of detection from XANES and EXAFS used in this work? Are they suitable for detecting surficial changes without being affected by bulk information?

Reply 1: Thanks for your careful reviewing. The Ni XANES and EXAFS profiles in this work were detected from Shanghai Synchrotron Radiation Facility with hard X-ray and the energy range of 4000 ~ 23000 eV. Such a high energy could be easily penetrating the Ni-rich materials, thus the XANES and EXAFS profiles revealed the overall element valence state, which has been widely established in previous literatures (*e.g.*, Xin *et al. Nature* 2022, 6, 610; Lu *et al. Nat. Commun.* 2022, 13, 2319).

Reply 2: Based on above analysis, as described in the original Manuscript (page 16, line 12), the EXAFS and EXAFS results (Fig. 5a-d) illuminated the overall valence of Ni element and the interaction between Ni and other elements (O, Co, Mn) after repeat cycles at fully charged state. However, due to the heterogeneous reaction inside electrode particles and the connection between various failure mechanisms, multi-scale (from surface to bulk) characterizations should be simultaneously carried out. Therefore, the s-XAS spectra with TEY mode (with a detection depth ~ 15 nm) have been employed (Fig. 1h, Supplementary Fig. 9-11, Supplementary Fig. 13) to elucidated the near-surface coordination environment. Furthermore, the TOF-SIMS (Fig. 4a-f) and HAADF-STEM (Fig. 5g-l) characterizations were also applied to illuminate the surface and the surface-to-bulk chemical environments in this work. As a result, combined the series of characterizations and theoretical simulations, the mechanism of LMO modifications on Ni-rich

material could be effectively elucidated. To clarify the XANES and EXAFS were revealed the overall valence state of Ni element, the corresponding description has been carefully clarified in the revised Manuscript.

Origin: (On page 16 in the original Manuscript)

Therefore, X-ray absorption near edge structure (XANES) spectra were collected to investigate the overall Ni valence state and the coordination environment (Fig. 5a, b).

Revision: (On page 16 in the revised Manuscript)

X-ray absorption near edge structure (XANES) technique with a high energy hard X-ray could be facilely penetrating the Ni-rich materials²⁸, thus was employed to investigate the overall Ni valence state and the coordination environment (Fig. 5a, b).

Addition: (On page 26 in the revised Manuscript)

28 Li, S. et al. Mutual modulation between surface chemistry and bulk microstructure within secondary particles of nickel-rich layered oxides. *Nat. Commun.* **11**, 4433 (2020).

11. Figure 5i. Where is the LMO deposition layer which is supposed to be on the NCM surface with the thickness at around 5 nm as indicated in Fig. 1c?

Reply 1: Thanks for your careful reviewing. The accelerating voltage applied for HR-TEM test in Fig. 1c was ~ 200 kV (voltage range for this device are 20 - 200 kV), whereas for HAADF-STEM in Fig. 5i was ~ 280 kV (voltage range: 60 - 300 kV). Therefore, the almost disappeared LMO layer in NCM surface (Fig. 5i) might be attributed to the high energy electrons bombardment, resulting in the chemical bonds breakage between the bulk NCM and LMO layer (e.g., Mary et al. *Sci. Adv.* 2023, 9, 3285; Nie et al. *Nat. Commun.* 2019, 10, 3243). To obtain the LMO layer involved in L-NCM HAADF-STEM image, the accelerating voltage was decreased to 200 kV, the result shown in the revised Fig. 5i clearly demonstrated that the LMO was still deposited on the cycled L-NCM surface. For consistency, the HAADF-STEM image for cycled P-NCM was also carried out with an accelerating voltage of 200 kV, and the corresponding images and descriptions have been strengthened in the revised Manuscript.

Origin: (On page 15 in the revised Manuscript)

Revision: (On page 15 in the revised Manuscript)

Addition 1: (On page 19 in the revised Manuscript)

Furthermore, the intact LMO shielding layer with a *P123* space group were tightly attached to its surface and proved by FFT³².

Addition 2: (On page 26 in the revised Manuscript)

32 Goutenoire, F. et al. Crystal structure of $\text{La}_2\text{Mo}_2\text{O}_9$, a new fast oxide-ion conductor. *Chem. Mater.* **12**, 2575 (2000).

Reviewer #3:

Reply: Thanks for your valuable advice and professional comment to help us improving the manuscript. Your suggestions and comments have been carefully considered and addressed, and all the responses are listed in the following section, point by point. In addition, all changes in the revised Manuscript have been highlighted in yellow for your convenience.

1) The manuscript reports an importance class of cathode materials in the field of Lithium-ion battery, with detailed and very insightful experimental and computational investigations. The results show clearly that the use of lanthanum molybdate improves the properties of P-NCM. However, it is necessary to compare these results with those similar-focused studies reported in the literature, especially experimental results, in order to better appreciate the importance of the present study. The following literature can help you: <https://doi.org/10.1016/j.ensm.2021.06.018>, <https://doi.org/10.1038/s41560-021-00780-2>, <https://doi.org/10.1016/j.mattod.2020.01.019>.

Reply: Thanks for your positive comment. As the reviewer suggested, the electrochemical performance of L-NCM in the present study have been compared with the reviewer suggested and the similar-focused studies to highlight the importance, which was found superior than the reported literatures. For clearly, the corresponding description and comparison were complemented, and the reviewer suggested as well as the similar-focused literatures have been added in the revised Manuscript and *Supplementary Information*.

Addition 1: (On page 10 in the revised Manuscript)

and was found superior compared with the recent similar-focused studies (Supplementary Table 4).

Addition 2: (On page 35 in the revised *Supplementary Information*)

Supplementary Table 4. Comparison in electrochemical performances of L-NCM on different voltages with the recent similar-focused literatures.

Bulk cathode	Modification strategies	Voltage range	Cycle number	Capacity retention	Reference
$\text{Li}_{0.8}\text{Ni}_{0.1}\text{Co}_{0.1}\text{Mn}_{0.1}\text{O}_2$	$\text{La}_2\text{Mo}_2\text{O}_9$ -coating	2.7-4.3 V	100	~ 93.9%	This work

$\text{Li}_{0.8}\text{Ni}_{0.1}\text{Co}_{0.1}\text{Mn}_{0.1}\text{O}_2$	$\text{La}_2\text{Mo}_2\text{O}_9$ -coating	2.7-4.4 V	100	~ 87.2%	This work
$\text{Li}_{0.8}\text{Ni}_{0.1}\text{Co}_{0.1}\text{Mn}_{0.1}\text{O}_2$	$\text{La}_2\text{Mo}_2\text{O}_9$ -coating	2.7-4.5 V	100	~ 84.8%	This work
$\text{Li}_{0.8}\text{Ni}_{0.1}\text{Co}_{0.1}\text{Mn}_{0.1}\text{O}_2$	DMS-electrolyte additive	2.75-4.5 V	100	~ 73.5%	10
$\text{LiNi}_{0.885}\text{Co}_{0.1}\text{Al}_{0.015}\text{O}_2$	B-doping	2.7-4.3 V	100	~ 95.5%	11
$\text{Li}_{0.8}\text{Ni}_{0.1}\text{Co}_{0.1}\text{Mn}_{0.1}\text{O}_2$	Sulfonamide-electrolyte additive	3.0-4.7 V	90	~ 88%	12
$\text{Li}_{0.8}\text{Ni}_{0.1}\text{Co}_{0.1}\text{Mn}_{0.1}\text{O}_2$	CoxB-coating	3.0-4.4 V	100	~ 87%	13
$\text{Li}_{0.8}\text{Ni}_{0.1}\text{Co}_{0.1}\text{Mn}_{0.1}\text{O}_2$	Element gradient distribution	2.7-4.4 V	100	~ 85.1%	14
$\text{Li}_{0.6}\text{Ni}_{0.2}\text{Co}_{0.2}\text{Mn}_{0.2}\text{O}_2$	Annealing in steps	3.0-4.3 V	100	~ 83%	15
$\text{Li}_{0.6}\text{Ni}_{0.2}\text{Co}_{0.2}\text{Mn}_{0.2}\text{O}_2$	Single-crystal	3.0-4.3 V	100	~ 89.1%	16
$\text{Li}_{0.76}\text{Ni}_{0.2}\text{Co}_{0.14}\text{Mn}_{0.1}\text{O}_2$	Single-crystal	3.0-4.3 V	100	~ 83.6%	17
$\text{Li}_{0.90}\text{Ni}_{0.2}\text{Co}_{0.09}\text{Mn}_{0.01}\text{O}_2$	Ta-doping	2.7-4.3 V	100	~ 75%	18

Addition 3: (On page 37 in the revised *Supplementary Information*)

10 Wu, F. et al. High-voltage and high-safety nickel-rich layered cathode enabled by a self-reconstructive cathode/electrolyte interphase layer. *Energy Storage. Mater.* **41**, 495 (2021).

11 Ryu, H. et al. A highly stabilized Ni-rich NCA cathode for high-energy lithium-ion batteries. *Mater. Today* **36**, 73 (2020).

12 Xue, W. et al. Ultra-high-voltage Ni-rich layered cathodes in practical Li metal batteries enabled by a sulfonamide-based electrolyte. *Nat. Energy* **6**, 495 (2021).

13 Ryu, H. et al. Reactive boride infusion stabilizes Ni-rich cathodes for lithium-ion batteries. *Nat. Energy* **6**, 362 (2022).

14 Lin, R. et al. Hierarchical nickel valence gradient stabilizes high-nickel content layered cathode materials. *Nat. Commun.* **12**, 2350 (2021).

- 15 Park, H. et al. In situ multiscale probing of the synthesis of a Ni-rich layered oxide cathode reveals reaction heterogeneity driven by competing kinetic pathways. *Nat. Chem.* **14**, 614 (2022).
- 16 Zhang, F. et al. Surface regulation enables high stability of single-crystal lithium-ion cathodes at high voltage. *Nat. Commun.* **11**, 3050 (2020).
- 17 Bi, Y. et al. Reversible planar gliding and microcracking in a single-crystalline Ni-rich cathode. *Science* **370**, 1313 (2020).
- 18 Sun, H. et al. Transition metal-doped Ni-rich layered cathode materials for durable Li-ion batteries. *Nat. Commun.* **12**, 6552 (2021).

2) A relationship must be established between experimental and theoretical results. Do the theoretical results agree with the experimental ones? If yes, specify in the manuscript.

Reply: Thanks for your professional comment. The liberation of oxygen in layered cathodes has been believed initiate at the surface and would react speedily with electrolyte, which not only resulted in the seriously electrochemical degradation on materials level, but also raised concerns about the thermal safety on commercial application. Therefore, defective oxygen inert phase $\text{La}_2\text{Mo}_2\text{O}_9$ (LMO) was carefully selected to engineer the $\text{LiNi}_{0.8}\text{Co}_{0.1}\text{Mn}_{0.1}\text{O}_2$ (NCM811) in this work. In this case, we proposed that the strong La-O and Mo-O bond lowered the $2p$ energy states of coordinated oxygen and thus served as “oxygen anchor” to alleviate the material failure caused by oxygen release originate from the surface, and the β -phase LMO layer with 41% oxygen vacancies could be in-situ reserved the possible released oxygen to ameliorate the exothermic reaction with electrolyte. As a result, the as-prepared L-NCM electrode exhibited almost no oxygen release at first and 200 cycles (Fig. 1f, g and Supplementary Fig. 7) with a high charge voltage (4.5 V) and excellent electrochemical performance.

Particularly, theoretical analysis was employed to confirm the oxygen anchoring and reserving process. As shown in Fig. 6c, d, due to the high binding energy of La-O and Mo-O compared with Ni-O, the energy states of surface O coordinated with La and Mo were decreased, thus contributing less high-energy states approach to Fermi level E_f than that of exposed surface oxygen (Fig. 6b) and lattice oxygen (Fig. 6a). The lowered energy levels of surface oxygen after LMO deposition could be attributed to the charge transfer from La and Mo ions, validly elevated

the electronegativities and would be demonstrated by charge density distributions (Fig. 6e, f) and Bader charge analysis (Supplementary Table 4), which greatly stabilized the surface oxygen (anchoring).

Furthermore, when oxygen vacancy was formed on LiNiO₂ (104) surface (Fig. 6g, h), an electronic state composed of Ni 3d and O 2p was introduced near the bottom of conduction band, and the calculated oxygen vacancy formation energy was ~ 3.1 eV. In contrast, when oxygen vacancy was existed on the LMO (001) surface (Fig. 6i, j), an intermediate electronic state composed La and Mo 4d orbital was introduced between the conduction band and the valence band. The oxygen vacancy formation energy of LMO was ~ 5.1 eV, which is about 2 eV higher than the LiNiO₂, indicating that the generation of O vacancy was more difficult in LMO. Therefore, LMO shielding layer would effectively reserve those oxygen species which were not anchored by La and Mo.

Therefore, the theoretical results obtained in Fig. 6 were agreed with the experimental and it has been discussed in the original Manuscript. To clarify more clearly, the corresponding description have been strengthened and specified in the revised Manuscript.

Origin: (On page 20 in the original Manuscript)

which greatly stabilized the surface oxygen (anchoring).

Revision: (On page 20 in the revised Manuscript)

which greatly stabilized the surface oxygen, and thus could be regarded as the “oxygen anchor” for Ni-rich cathode.

Addition: (On page 20 in the original Manuscript)

To illustrate the “oxygen reserving” effect of LMO layer, the intrinsic oxygen vacancy formation energy of LNO (104) and LMO (001) surface were calculated and discussed (Fig. 6g, h and Supplementary Fig. 25a, b).

3) Please the quality of your image should be improved.

Reply: Thanks for your kind reminding. The pixel of those images in the original Manuscript were 300 dots per inch (dpi). To quality the images, the corresponding dpi were set as 600 (the highest dpi in Origin 2023) and the detail of the images have been carefully revised (**The revisions have been presented in the revised Manuscript and Supplementary Information**). Furthermore,

according to the demand of *Nature Communications* editorial office, if this Manuscript could be accepted, the highly-quality vector images should be submitting individually for the final publications.

4) The slight challenges I encountered during the review has to do with the language. The English of the manuscript needs extensive revision.

Reply: Thanks for your constructive suggestion. As the reviewer suggested, the English of this Manuscript has been carefully revised and highlighted in yellow color for your convenience (**The revisions have been presented in the revised Manuscript**).

REVIEWERS' COMMENTS

Reviewer #2 (Remarks to the Author):

I believe the authors did a great job revising the manuscript. The additional experiments and discussions provided thorough answers to my comments. The quality of this manuscript was dramatically improved by the careful revision from the authors. I don't have any further questions. I recommend the publication of this revised manuscript in Nature Communications.

Reviewer #4 (Remarks to the Author):

I suppose the author has done a great job. I feel like it can be accepted directly. The responses have appropriately addressed those concerns from referee 1.

REVIEWERS' COMMENTS

Reviewer #2 (Remarks to the Author):

I believe the authors did a great job revising the manuscript. The additional experiments and discussions provided thorough answers to my comments. The quality of this manuscript was dramatically improved by the careful revision from the authors. I don't have any further questions. I recommend the publication of this revised manuscript in Nature Communications.

Reviewer #4 (Remarks to the Author):

I suppose the author has done a great job. I feel like it can be accepted directly. The responses have appropriately addressed those concerns from referee 1.

Detailed Responses to Reviewers' Comments

Reviewer #2:

General comment: I believe the authors did a great job revising the manuscript. The additional experiments and discussions provided thorough answers to my comments. The quality of this manuscript was dramatically improved by the careful revision from the authors. I don't have any further questions. I recommend the publication of this revised manuscript in Nature Communications.

Reply: We thank reviewer for the "don't have any further questions and publication" suggestion.

Reviewer #4:

General comment: I suppose the author has done a great job. I feel like it can be accepted directly. The responses have appropriately addressed those concerns from referee 1.

Reply: We thank reviewer for the "accepted directly" suggestion.